# High allelic diversity in *Arabidopsis* NLRs is associated with distinct genomic features

Chandler A Sutherland [ID][1], Daniil M Prigozhin[2], J Grey Monroe [ID][3] & Ksenia V Krasileva [ID][1✉]

## Abstract

**Plants rely on Nucleotide-binding, Leucine-rich repeat Receptors (NLRs) for pathogen recognition. Highly variable NLRs (hvNLRs) show remarkable intraspecies diversity, while their low-variability paralogs (non-hvNLRs) are conserved between ecotypes. At a population level, hvNLRs provide new pathogen-recognition specificities, but the association between allelic diversity and genomic and epigenomic features has not been established. Our investigation of NLRs in *Arabidopsis* Col-0 has revealed that hvNLRs show higher expression, less gene body cytosine methylation, and closer proximity to transposable elements than non-hvNLRs. hvNLRs show elevated synonymous and nonsynonymous nucleotide diversity and are in chromatin states associated with an increased probability of mutation. Diversifying selection maintains variability at a subset of codons of hvNLRs, while purifying selection maintains conservation at non-hvNLRs. How these features are established and maintained, and whether they contribute to the observed diversity of hvNLRs is key to understanding the evolution of plant innate immune receptors.**

**Keywords** Immunity; Nucleotide-binding Leucine-rich-repeat Receptors (NLRs); Genomic Features; Evolution
**Subject Categories** Chromatin, Transcription & Genomics; Evolution & Ecology; Immunology

## Introduction

Plants, lacking the adaptive immune systems of vertebrates, use germline-encoded innate immune receptors to defend against rapidly evolving pathogens. Despite their inability to create antibodies through somatic hypermutation and recombination, plants are protected against pathogens due to population-level receptor diversity (Bakker et al, 2006; Van de Weyer et al, 2019; Karasov et al, 2020; Kahlon and Stam, 2021). Nucleotide-binding, Leucine-rich repeat Receptors (NLRs) are the intracellular sensors of the plant immune system, detecting pathogen-secreted, disease-promoting effector proteins (Jones et al, 2016). NLRs have a modular domain structure, with a variable N-terminal domain

involved in downstream signaling, a central nucleotide-binding domain shared by APAF-1, various other plant immune proteins, and CED4 (NBARC), and a leucine-rich repeat (LRR) domain involved in direct or indirect recognition of pathogens (Ngou et al, 2022). NLRs are grouped into three anciently diverged classes based on their N-terminal domains: coiled-coil (CC) NLRs (CNL), RPW8-like coiled-coil NLRs (RNL), and Toll/Interleukin-1 receptor (TIR) NLRs (TNLs) (Shao et al, 2016; Tamborski and Krasileva, 2020). After pathogen recognition, NLRs initiate defense responses through oligomerization of the NBARC domain, leading to transcriptional reprogramming, hormone induction, and hypersensitive cell death response (Ngou et al, 2022).

NLRs exhibit remarkable levels of intraspecies allelic diversity (Van de Weyer et al, 2019), due to both the genomic processes that generate variation and selection that promotes its maintenance (Karasov et al, 2014a; Barragan and Weigel, 2021; Märkle et al, 2022). NLRs are in close proximity to each other in genomes and are organized into clusters more often than other genes. This proximity can asymmetrically drive NLR expansion and diversification through tandem duplication, unequal crossing over, and gene conversion (Parker et al, 1997; Michelmore and Meyers, 1998; Lee and Chae, 2020) as well as accumulation of point mutations (Kuang et al, 2004). Point mutations are a major source of within-species NLR diversity but have been difficult to fully resolve through short-read sequencing approaches. The NLR gene family includes the most polymorphic loci and contains the highest frequency of major effect mutations in the *Arabidopsis* genome (Clark et al, 2007; Gan et al, 2011). There is evidence for balancing selection maintaining polymorphisms and presence-absence variation at several NLR loci through frequency-dependent selection, spatial and temporal fluctuations in pathogen pressure, and heterozygote advantage (Thrall et al, 2012; Karasov et al, 2014b; MacQueen et al, 2019). Diversifying selection has also been observed at NLR loci as an excess of nonsynonymous to synonymous substitutions (Bakker et al, 2006). The NLR gene and protein sequences within a species represent a snapshot of the ongoing interplay between mutation and selection, but disentangling their relative contributions remains challenging.

Mutation rates are unlikely to evolve on a gene-by-gene basis in response to selection given the barrier imposed by genetic drift (Lynch, 2010). However, selection on genic mutation rates is sufficiently strong when acting on mechanisms that couple mutation rate to expression states and epigenomic features, affecting the mutation rates of many genes simultaneously

[1]Department of Plant and Microbial Biology, University of California Berkeley, Berkeley, CA 94720, USA. [2]Molecular Biophysics and Integrated Bioimaging Division, Lawrence Berkeley National Laboratory, Berkeley, CA 94720, USA. [3]Department of Plant Sciences, University of California Davis, Davis, CA 95616, USA. ✉E-mail: kseniak@berkeley.edu

(Martincorena and Luscombe, 2013). The mutation rate of *Arabidopsis* is heterogeneous across the genome, consistent with the expected effects of selection on mechanisms linking mutation rates to epigenomic features (Monroe et al, 2022a; Staunton et al, 2023). Several mechanisms have been described, including cytosine methylation, which is positively correlated with mutation probability and known to increase the likelihood of spontaneous deamination (Cao et al, 2011; Weng et al, 2019) and H3K4me1, which is negatively correlated with mutation probability and a target of several DNA repair proteins (preprint: Quiroz et al, 2022). Description of genomic features associated with diversity in NLRs will help to understand the role of mutation bias in NLR evolution.

Recent advances in enrichment-based long-read sequencing of NLRs (Jupe et al, 2013) as well as long-read pan-genomes (Jiao and Schneeberger, 2020) allowed for re-examination of NLR variation within species (Barragan and Weigel, 2021). In *Arabidopsis* datasets, it has been shown that NLRs are enriched in regions of synteny diversity and that NLR repertoires across species could not be easily anchored to a reference genome (Van de Weyer et al, 2019). Phylogenetic analysis independent of reference-based assignment of pan-NLRomes from 62 *Arabidopsis thaliana* accessions (Van de Weyer et al, 2019) and 54 *Brachypodium distachyon* (Gordon et al, 2017) lines allowed for amino acid diversity quantification and delineation of highly variable NLRs (hvNLRs) from their low-variability paralogs (non-hvNLRs) (Prigozhin and Krasileva, 2021). At the species level, hvNLRs show rapid rates of diversification and are hypothesized to act as reservoirs of diversity for recognition of pathogen effectors. Comparison of hv and non-hvNLR gene sets allows for investigation of epigenomic, sequence, and regulatory features (hereafter genomic features) and signatures of selection associated with NLR diversification.

In this paper, we report that hvNLRs show a higher transcription level, less gene body CG methylation, and closer proximity to transposable elements (TEs) than non-hvNLRs. Elevated gene-wide nucleotide diversity, a higher likelihood of mutation, and diversifying selection at a subset of sites promote the high amino acid diversity of hvNLRs, while non-hvNLRs are subject to purifying selection. These findings will serve as a starting point for the investigation of the mechanisms that generate and maintain diversity in a subset of plant immune receptors.

## Results

### Shannon entropy delineates highly variable NLRs in *Arabidopsis* Col-0

Shannon entropy, a measure of variability derived from information theory, provides an unbiased metric of amino acid diversity of a protein within a population (Asti et al, 2016; Wang et al, 2017). Here, the Shannon entropy is the sum of the frequency of each amino acid times the logarithm of that frequency at each position in a protein sequence alignment, so sites with low variability have low entropy and highly diverse sites have high entropy. When applied to NLRs, this measure is predictive of highly variable effector binding sites (Prigozhin and Krasileva, 2021) and can be used in NLR ligand binding site engineering (Tamborski et al, 2023). Based on the bimodal distribution of Shannon entropy in

NLRome, hvNLRs are defined as proteins with 10 or more amino acid positions with Shannon entropy greater than 1.5 bits (Fig. EV1) (Prigozhin and Krasileva, 2021). To examine the relationships between population-level diversity and genomic features of a single accession, we plotted Shannon entropy in reference to each NLR in Col-0 (Fig. 1). As expected, there are functional hvNLRs and non-hvNLRs, with known direct recognition of effectors corresponding to hvNLRs and known indirect recognition to non-hvNLRs. In addition, hvNLRs include all currently known dangerous mix genes that are responsible for hybrid incompatibility across *Arabidopsis* accessions (Bomblies et al, 2007; Chae et al, 2014). Categorizing NLRs into low and high-entropy groups allows for pairwise comparison of features between groups and gene set enrichment analysis to compare NLRs to the rest of the genome.

### hvNLRs have distinct genomic features from non-hvNLRs and the rest of the genome

To compare the expression and methylation status of hv and non-hvNLR gene sets within an individual plant, we examined available paired whole genome bisulfite and RNA sequencing generated from the same rosette leaf of the reference accession Col-0 (Data ref: Williams et al, 2022a, 2022b). We found that the distribution of hvNLR expression is significantly higher than non-hvNLRs (Fig. 2A, unpaired Wilcoxon rank-sum test, $P = 7.9e-05$). When we ranked all protein-coding *Arabidopsis* genes based on their expression level, we observed that hvNLRs are enriched in the most expressed genes in each leaf sample (Fig. 2D, singscore rank-based sample scoring, $P < 0.005$ for hvNLRs in each biological replicate).

In addition, the hvNLR gene set is significantly less CG gene body methylated than non-hvNLRs (Fig. 2B, unpaired Wilcoxon rank-sum test, $P = 1.2e-04$). We noticed two hvNLRs, *RPP4* and *RPP7*, with higher CG methylation than the average for hvNLRs (Fig. 2B). Upon further inspection, we also found CHH and CHG context methylation within the gene bodies of *RPP4* and *RPP7*, which we rarely observed in other NLRs (Fig. EV2A–C; median

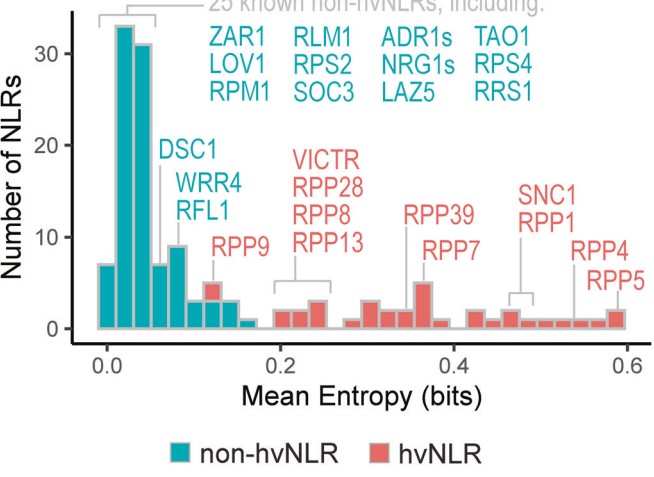

**Figure 1. hvNLRs are defined by high amino acid diversity.**

Distribution of mean Shannon entropy per gene calculated in reference to Col-0. NLRs shown by a histogram with 30 bins. Named NLRs with previous functional characterization are labeled. Source data are available online for this figure.

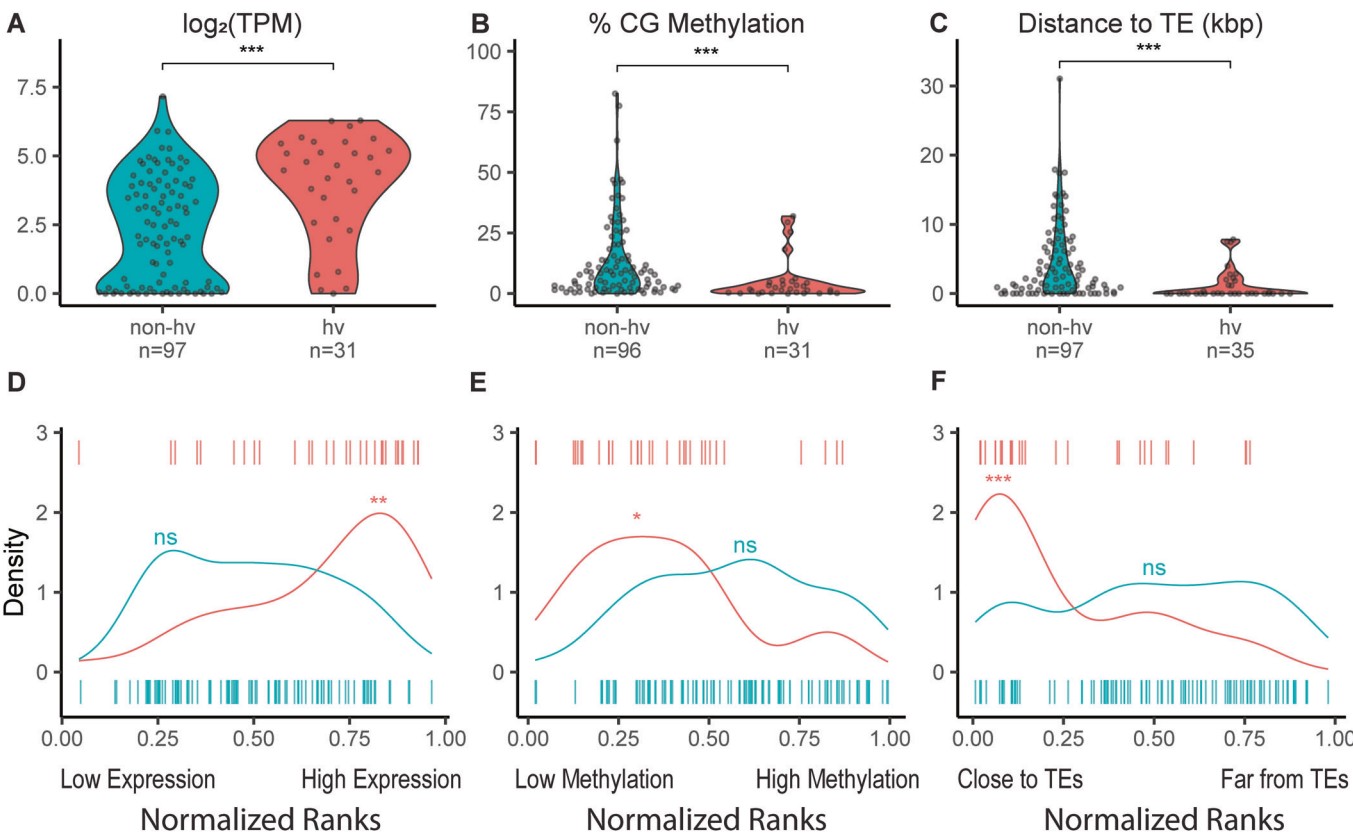

**Figure 2. Expression, methylation, and proximity to transposable elements (TEs) distinguish hv and non-hvNLRs in Col-0 rosette tissue.**

(A–C) Average gene expression log₂ (transcripts per million (TPM)), average % CG methylation per gene, and distance to the nearest TE (kbp) of hv and non-hvNLRs. (D–F) Normalized mean percentile rank density plots of hv and non-hvNLRs. Data Information: (A–C) significance shown is the result of unpaired Wilcoxon rank-sum tests with Benjamini–Hochberg correction for multiple testing. (D–F) The significance shown is the result of singscore rank-based sample scoring or permutation test for difference in means. For description of the statistical test, see "Methods". "ns" indicates a $P$ value > 0.05; * indicates a $P$ value < 0.05 and ≥0.01; ** indicates a $P$ value < 0.01 and ≥0.001; *** indicates a $P$ value < 0.001. $n$ refers to the number of NLR genes tested. Source data are available online for this figure.

NLR CHH methylation = 0.29%, median NLR CHG methylation = 0.30%). Multi-context gene body methylation (CG, CHH, and CHG) is typically used to silence nearby or overlapping transposable elements (Quadrana et al, 2016). This indicates that the elevated CG methylation of *RPP4* and *RPP7* is likely due to multi-context silencing related to a recent TE insertion, which has been previously reported (Tsuchiya and Eulgem, 2013). hvNLRs are enriched in the CG hypomethylated genes across the genome (permutation test for difference in means, $P = 0.002$, $n = 10,000$ replicates). Gene set analysis of methylation can be biased due to the uneven distribution of CG sites within each gene (Geeleher et al, 2013). We repeated our permutation test to compare hvNLRs to a set of non-NLR genes with similar measured CG sites per gene to correct for this bias, and hvNLRs were significantly more hypomethylated than the rest of the genome (Fig. 2E, $P < 0.05$ for each biological replicate, $n = 10,000$).

We also found that hvNLRs as a set are closer to TEs (Fig. 2C, unpaired Wilcoxon rank-sum test, $P = 1.3e-05$), and hvNLRs are enriched in the genes closest to TEs (Fig. 2F, permutation test for difference in medians, $P = 0$, $n = 10,000$ replicates). In Col-0, hvNLRs have a median TE distance of 0 kbp, meaning the TEs are within the UTR or intronic sequences, while non-hvNLRs have a median TE distance of 2.07 kbp. The highly variable status of NLRs

is predictive of TEs within the genic sequence (Fisher's exact test, $P = 3.6e-05$). It has been previously observed that TEs are associated with plant immune genes (Hosaka and Kakutani, 2018), but this analysis suggests that the signal is driven by hvNLRs.

## Genomic features are robust to cluster status, domain class, and phylogenetic distance

NLRs are found in clusters more frequently than other genes, with clusters defined as a maximum distance of 50 kb between adjacent NLRs (Lee and Chae, 2020), and NLRs belong to three anciently diverged classes named for their N-terminal domains (Fig. 3B; Shao et al, 2016; Tamborski and Krasileva, 2020). While there is not a significant association between the highly variable status of NLRs with cluster membership (Fisher's exact test, $P = 1$) or N-terminal domain clade (Fisher's exact test, $P = 0.21$), we wanted to examine both factors as potential confounders of our feature associations. We therefore subset our data by cluster status and N-terminal domain and repeated hv and non-hvNLR pairwise comparisons (Fig. 3A) and examined the distribution of features across the phylogeny of Col-0 NLRs (Fig. 3B). The distribution of hvNLR expression is significantly higher than non-hvNLRs, and

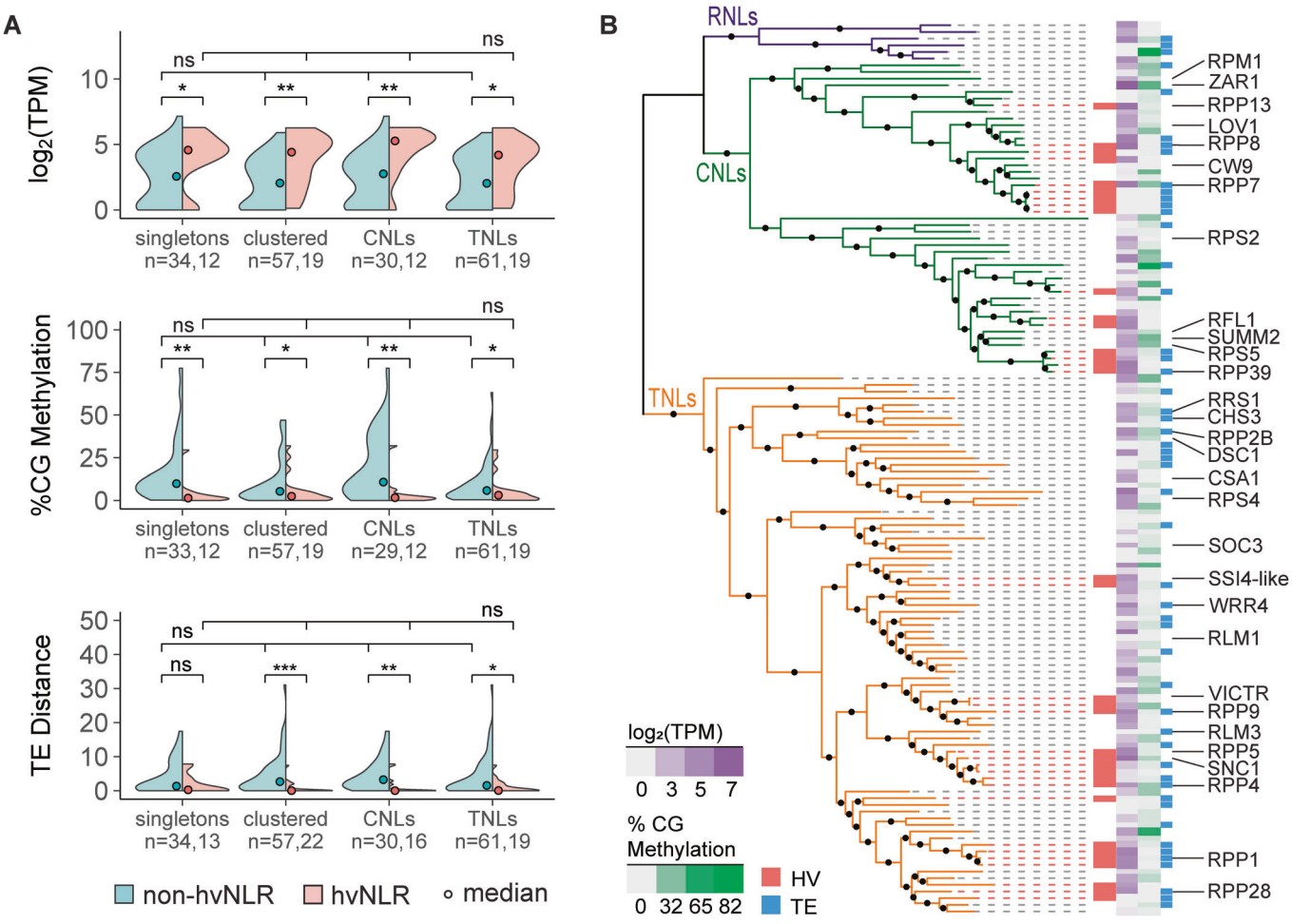

**Figure 3. Cluster membership, NLR type, and phylogenetic distance do not explain genomic differences between hv and non-hvNLRs.**

(A) Comparison of expression, distance to nearest TE, and CG gene body methylation of hv and non-hvNLRs subset by cluster membership and N-term domain type. (B) Features mapped onto a phylogeny of NLRs in *A. thaliana* Col-0. NLRs without log$_2$(TPM) or %CG methylation data were determined to be unmappable (see "Methods"). Named NLRs with previous functional characterization are labeled. Data Information: (A) pairwise significance shown is the result of an unpaired Wilcoxon rank-sum test with Benjamini–Hochberg correction for multiple testing. Across subset significance shown is the result of a Kruskal–Wallis test with Benjamini–Hochberg correction for multiple testing. "ns" indicates a *P* value > 0.05; * indicates a *P* value <0.05 and ≥0.01; ** indicates a *P* value < 0.01 and ≥0.001; *** indicates a *P* value < 0.001. *n* refers to the number of NLR genes tested in each subset. Source data are available online for this figure.

the distribution of hvNLR %CG methylation is significantly lower than non-hvNLRs within each subset (Fig. 3A, unpaired Wilcoxon rank-sum tests, corrected for multiple hypothesis testing). TE distance is significantly different in clustered NLRs, CNLs, and TNLs, but not singletons. hvNLRs are distributed over the phylogeny of CNLs and TNLs and maintain distinct genomic features despite close phylogenetic relationships with non-hvNLRs (Fig. 3B).

If cluster status or domain class were confounding variables, we would expect statistically different distributions across hvNLR subsets and non-hvNLR subsets. However, the distributions are not significantly different across all features for both hvNLRs and non-hvNLRs (Fig. 3A; Kruskal–Wallis rank-sum tests corrected for multiple hypothesis testing). This analysis shows that the differences observed between hvNLRs and non-hvNLRs are consistent across the NLR phylogeny and are not driven by cluster membership.

## hvNLRs are more expressed and less methylated than non-hvNLRs across tissues

Since our investigation of genomic features of NLRs is so far exclusive to rosette leaf tissue, we were curious if our observed trends held across other tissues. We employed an additional dataset that sampled RNA from 30 tissues of *Arabidopsis* Col-0 across several developmental stages for a total of 52 samples (Data ref: Mergner et al, 2020a, 2020b). As a gene set, hvNLRs are more highly expressed than non-hvNLRs in 46 of the 52 samples examined (Fig. 4A; unpaired Wilcoxon rank-sum tests corrected for multiple testing). We see the significantly greater expression of hvNLRs than non-hvNLRs in several reproductive tissues, including all sampled stages of flower development and silique development, fruit tissue, and the majority of embryo samples (Fig. 4A). As previously reported (Munch et al, 2018), we observe that NLRs are more expressed in vegetative tissues than

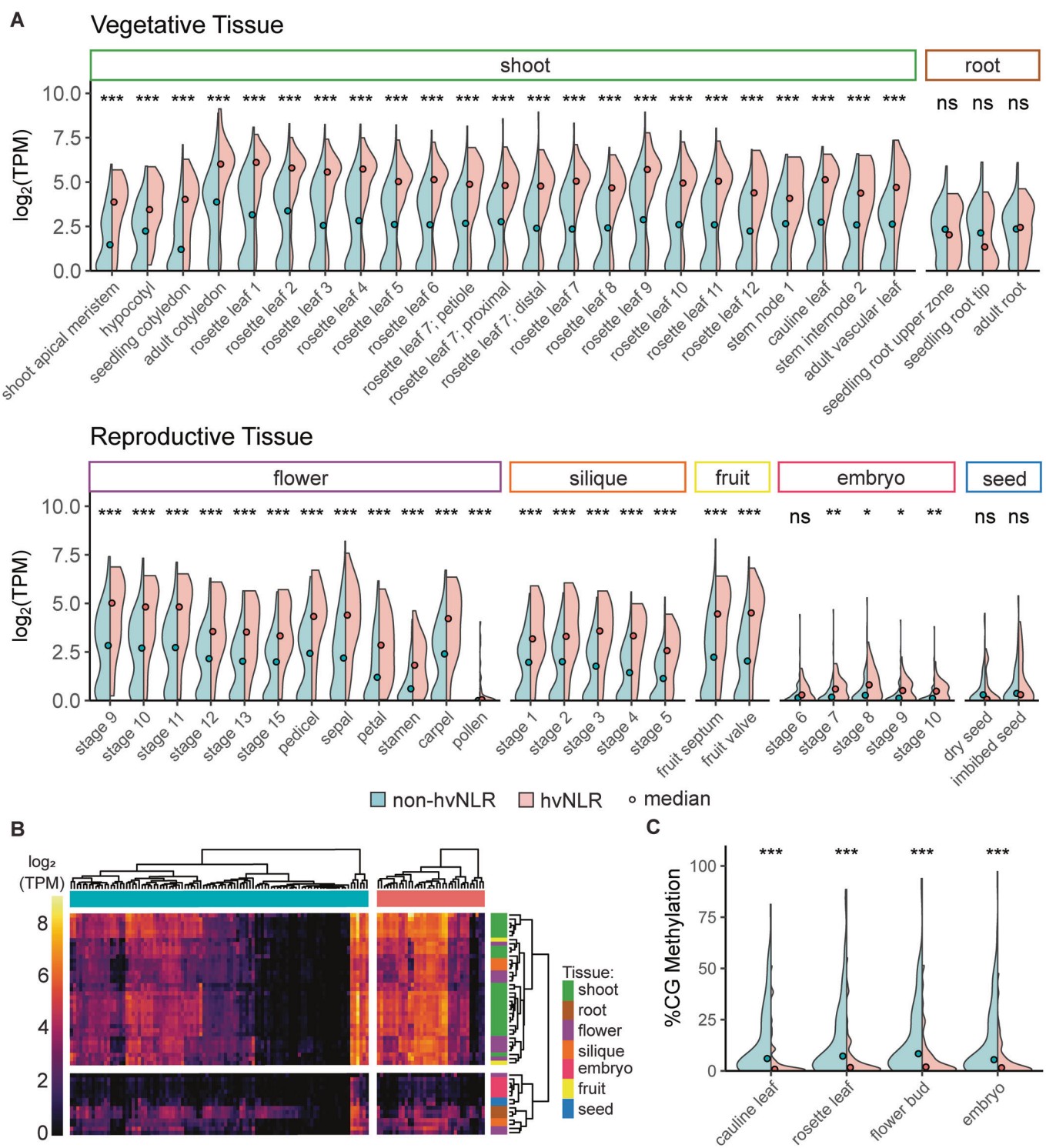

**Figure 4. Expression and methylation features are conserved across tissue types and developmental stages.**

(A) Pairwise comparison of hv vs non-hvNLR average gene expression in 52 tissue and developmental stage samples. Each comparison includes $n = 97$ non-hvNLRs and $n = 35$ hvNLRs, with $n$ referring to the number of genes tested. (B) Heatmap of hv and non-hvNLR expression in $\log_2$ (transcripts per million (TPM)). Dendrograms show the result of hierarchical clustering across tissues and within hv and non-hvNLR gene sets. (C) %CG gene body methylation of hv and non-hvNLRs in four tissue types. Each comparison shows $n = 95$ non-hvNLRs and $n = 32$ hvNLRs genes, with n referring to the number of genes tested. Data Information: (A, C) significance shown is the result of unpaired Wilcoxon rank-sum tests with Benjamini–Hochberg correction for multiple testing. "ns" indicates a $P$ value > 0.05; * indicates a $P$ value < 0.05 and ≥0.01; ** indicates a $P$ value < 0.01 and ≥0.001; *** indicates a $P$ value < 0.001. Source data are available online for this figure.

reproductive tissues, and more expressed in shoot tissue than root tissue (Fig. 4B). There is no significant difference between the two groups in embryo developmental stage 6, dry or imbibed seed tissue, or any of the three root tissues.

While CG methylation varies less between tissue types than expression (Seymour et al, 2014; Lloyd and Lister, 2022), we also compared hv and non-hvNLR methylation in cauline leaves, flower buds, embryos, and a distinct rosette leaf sample (Data ref: Williams et al, 2022a, 2022b). non-hvNLRs are more CG methylated than hvNLRs in all tissues examined (Fig. 4C, unpaired Wilcoxon rank-sum tests corrected for multiple testing). Therefore, we conclude that in Col-0, hv-associated genomic features are conserved across most tissue types, including in reproductive organs.

## non-hvNLRs are subject to purifying selection

The high level of amino acid diversity in hvNLRs and associated differences in genomic features might be due to differences in mutational processes and/or selection. To investigate the contribution of balancing selection to the observed amino acid diversity at hvNLRs, we calculated Tajima's D (D) and nucleotide diversity per site ($\pi$) in each domain and across the gene body of hv and non-hvNLRs. D is a site frequency spectrum-based statistic that tests for selection by comparing the difference between the average number of nucleotide differences and the total number of segregating sites to the neutral expectation, while $\pi$ measures the degree of polymorphism within a population by the average pairwise differences per site. In comparison

to the rest of the genome, these statistics can be used to test for balancing selection (Schmid et al, 2005). hvNLRs have higher D than non-hvNLRs across the coding sequence and all individual domains (Fig. 5A; unpaired Wilcoxon rank-sum test corrected for multiple testing). Reflecting their differences in amino acid diversity, hvNLRs have higher $\pi$ than non-hvNLRs across all domains and the coding sequence (Fig. 5A; unpaired Wilcoxon rank-sum test corrected for multiple testing). The difference in $\pi$ and D between the two groups is not driven exclusively by variation in the LRR region, with the highest values reported for the hvNLR NBARC domains.

Due to the demographic history of *Arabidopsis*, the empirical distribution of summary statistics departs from the neutral model (Nordborg et al, 2005; Schmid et al, 2005; Alonso-Blanco et al, 2016). We calculated the genome-wide values of D and $\pi$ to test for selection, using whole genome SNP information from the accessions used to create the pan-NLRome present in 1001 genomes (Fig. EV3A,B; Alonso-Blanco et al, 2016). Both hv and non-hvNLRs have higher average $\pi$ than the empirical distribution (Fig. 5B; permutation test for difference in means, $P = 0$; $P = 0$, $n = 10,000$ replicates), and there are significantly more NLRs in the top 5% of the empirical distribution than expected by chance (Fig. EV3B; permutation test for number in the top 5%, $P = 0$, $n = 10,000$ replicates). This corroborates previously reported significantly high levels of nucleotide diversity of NLRs (Bakker et al, 2006; Van de Weyer et al, 2019).

hvNLRs have a higher D, and non-hvNLRs have a lower D than the genome average (Figs. 5B and EV3A; permutation test for

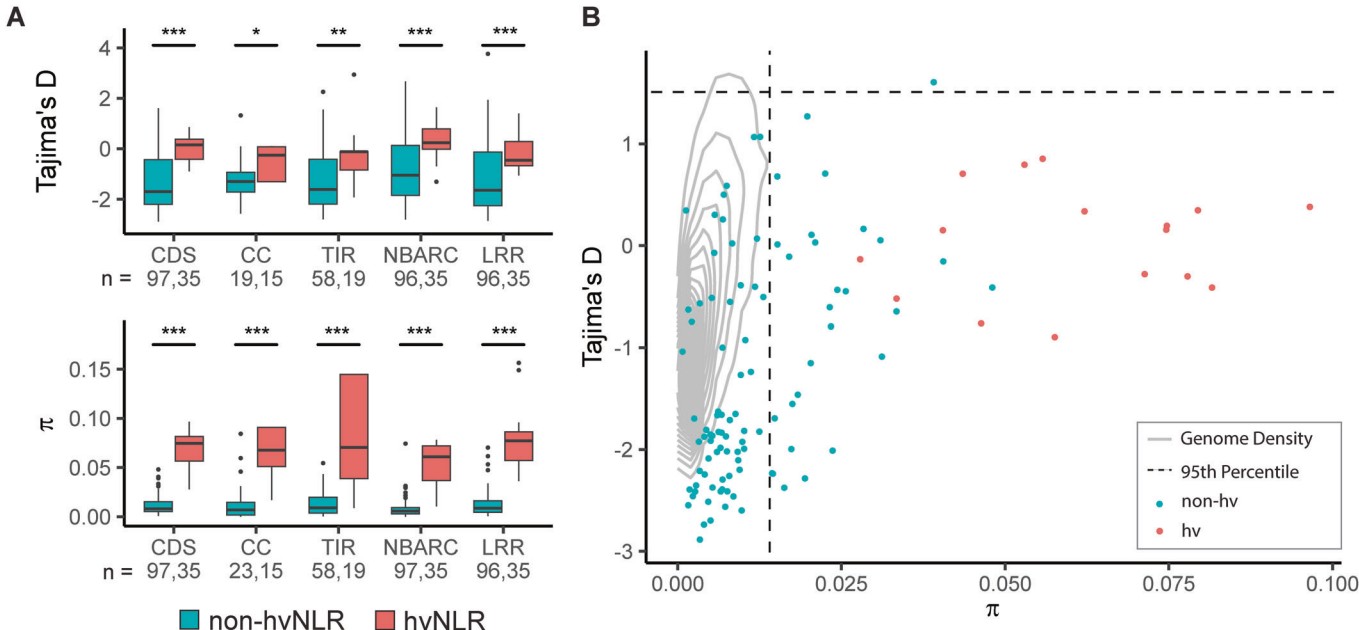

**Figure 5. hvNLRs have higher Tajima's D and nucleotide diversity than non-hvNLRs.**

(A) D and $\pi$ calculated across the coding sequence (CDS), coiled-coil (CC), Toll/Interleukin-1 (TIR), nucleotide-binding (NBARC), and leucine-rich repeat (LRR) domains. (B) CDS $\pi$ vs. D. Gray lines represent the kernel density estimation of statistics computed on all coding sequences of *Arabidopsis*. Dashed lines represent the 95th percentile of the empirical distribution. Data Information: (A) horizontal black lines denote median values within each box; boxes range from the 25th to 75th percentile of each group's distribution of values; whiskers extend no further than 1.5× the interquartile range of the hinge. Data beyond the end of the whiskers are outlying points and are plotted individually. *n* refers to the number of genes tested. Significance shown is the result of unpaired Wilcoxon rank-sum tests with Benjamini–Hochberg correction for multiple testing. * Indicates a *P* value < 0.05 and ≥0.01; ** indicates a *P* value < 0.01 and ≥0.001; *** indicates a *P* value < 0.001. Source data are available online for this figure.

difference in means, $P = 9.0e-04$; $P = 0$, $n = 10,000$ replicates). There are no hvNLRs in either tail of the empirical distribution of D, which is not significantly different from the 3.45 expected by chance. There is, however, an excess of non-hvNLRs in the bottom 5% of the distribution of D (permutation test for number in the bottom 5%, $P = 0$, $n = 10,000$ replicates), indicating that purifying selection may be reducing diversity at non-hvNLRs. Defining individual genes under balancing selection to be the top 5% of the empirical distribution of $\pi$ and D values (Bakker et al, 2006; preprint: Gladieux et al, 2022), we identified one non-hvNLR under balancing selection, AT5G47260 (Fig. 5B). However, one gene is not significantly different from the number of NLRs expected to be in the top 5% of both distributions by random chance.

## hvNLRs are subject to diversifying selection and an increased probability of mutation

To further investigate the nature of the high nucleotide diversity of NLRs, we compared nucleotide diversity at synonymous and nonsynonymous sites ($\pi_S$; $\pi_N$). hvNLRs have greater $\pi_S$ and $\pi_N$ than non-hvNLRs (Fig. 6A; unpaired Wilcoxon rank-sum tests corrected

for multiple testing $P = 1.2e-12$, $P = 3.3e-15$). However, the ratio of nonsynonymous to synonymous nucleotide diversity ($\pi_N/\pi_S$), an intraspecies measurement of selection, is not significantly different between the two groups, indicating possible role of different mutational processes (Fig. 6B; unpaired Wilcoxon rank-sum test, $P = 0.17$). Average $\pi_N/\pi_S$ is less than 1 for both hv and non-hvNLRs across the gene and in the LRR region, indicating purifying selection as an excess of synonymous polymorphisms relative to nonsynonymous polymorphisms (Figs. 6B and EV3C).

Since elevated $\pi_N$ and $\pi_S$ with no difference in $\pi_N/\pi_S$ could be caused by an increase in the mutation rate of hvNLRs, we compared the predicted SNVs and indels per base pair based on epigenomic states (mutation probability score) (Data ref: Monroe et al, 2022a, 2022b). The median mutation probability is 35% higher for hvNLRs (Fig. 6C; unpaired Wilcoxon rank-sum test, $P = 2.5e-05$).

Gene-wide $\pi_N/\pi_S$ is a conservative metric for testing positive selection because positive selection may only be acting at a few codon sites (Kosakovsky Pond and Frost, 2005). Therefore, we used maximum likelihood-based site models to test for positive, diversifying selection. The use of these dN/dS-based models on intraspecies data is problematic because the nucleotide differences do not represent

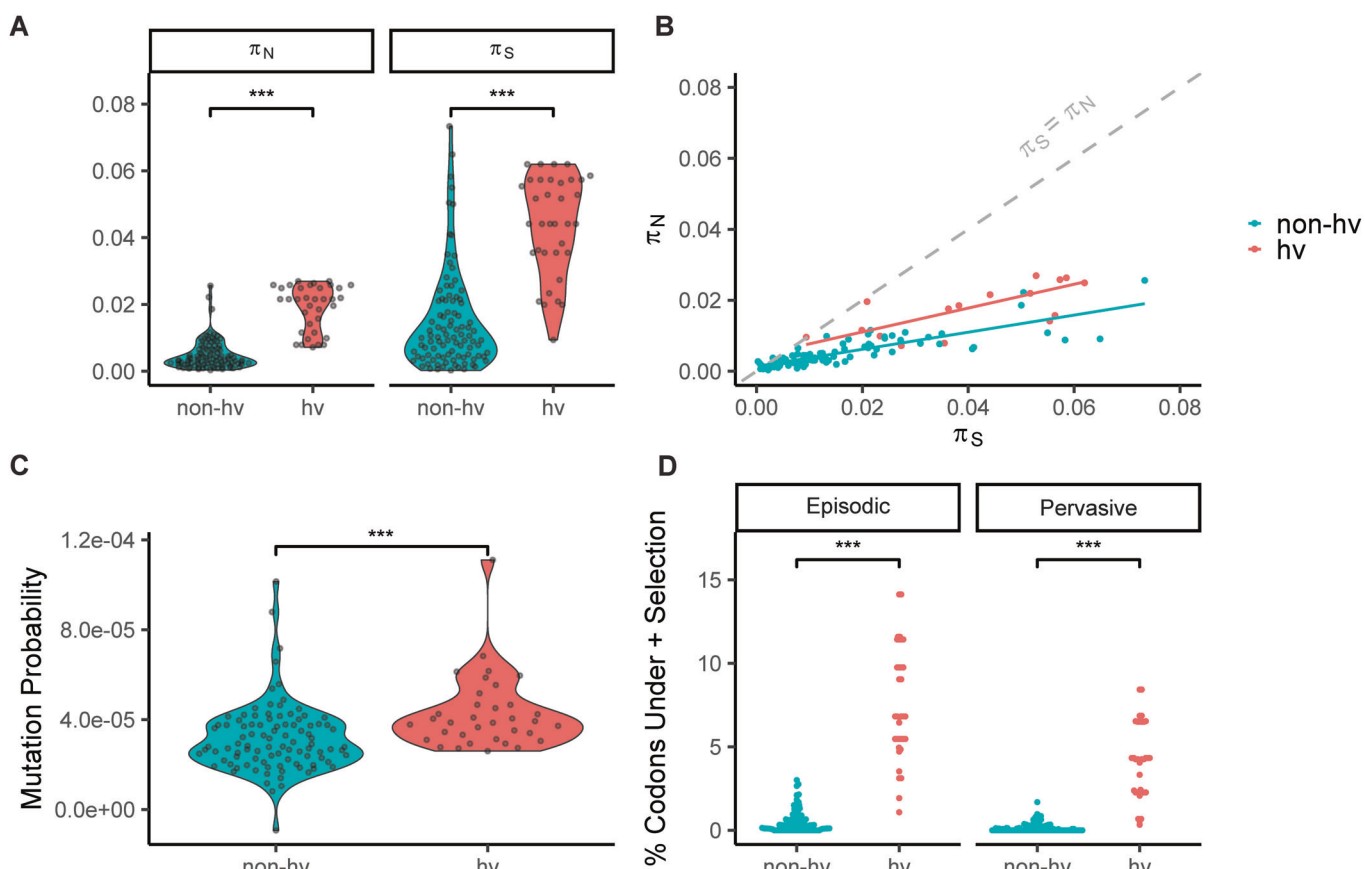

**Figure 6. hvNLR nucleotide diversity is associated with a high likelihood of mutation and codons under diversifying selection.**

(A) Average nonsynonymous pairwise nucleotide diversity per site ($\pi_N$) and average synonymous pairwise nucleotide diversity per site ($\pi_S$). (B) $\pi_S$ vs $\pi_N$ of the coding sequence of NLRs with per group linear regressions. The neutral expectation, $\pi_S = \pi_N$, is shown as a gray dotted line. (C) Mutation probability score of hv and non-hvNLRs. (D) Percentage of codons under positive selection determined by MEME (episodic), and FEL (pervasive). Data Information: For all comparisons, $n = 97$ for non-hvNLRs and $n = 35$ for hvNLRs, with $n$ referring to the number of genes tested. The significance shown is the result of unpaired Wilcoxon rank-sum tests, with *** indicating a $P$ value < 0.001. Source data are available online for this figure.

substitutions fixed by selection, but rather polymorphisms segregating within a population (Kryazhimskiy and Plotkin, 2008). We mitigated this effect by restricting our analysis to internal branches of the protein phylogeny, which encompass at least one ancestral sequence that is visible to selection (Pond et al, 2006; Avanzato et al, 2019). hvNLRs have a higher proportion of codons under pervasive and episodic diversifying selection than non-hvNLRs, indicating that diversifying selection at a subset of sites is maintaining diversity at hvNLRs (Fig. 6D, unpaired Wilcoxon rank-sum tests). Given the polymorphism data, summary statistics, and mutational likelihood, hvNLR amino acid diversity appears to be driven by both a higher likelihood of mutation and positive, diversifying selection, while non-hvNLR conservation is maintained by purifying selection.

### Distinct genomic features and signatures of selection can persist in close proximity

As described previously, hv and non-hvNLRs can co-exist within the same cluster and even as neighboring genes (Fig. EV2A,B). We

chose to examine the *RSG2* cluster, which includes neighboring non-hvNLR *RSG2* and hvNLR *AT5G43740*, two CNLs of similar length 1.8 kb apart, as a representative example of neighboring NLRs. The hvNLR is highly expressed, hypomethylated, and has a TE within its 5' UTR sequence (Fig. 7A). The non-hvNLR shows signatures of purifying selection with a gene-wide Tajima's D value of $-1.9$, while the hvNLR has a gene-wide Tajima's D of $-0.24$ (Fig. 7B,C). The hvNLR has higher $\pi$, $\pi_N$, and $\pi_S$, but the two genes have similar $\pi_N/\pi_S$ values (0.48 and 0.41) (Fig. 7B,C). Despite neighboring genomic positions, *RSG2* and *AT5G43740* show distinct genomic features and signatures of selection reflective of their species-level amino acid diversity. There are two other examples of neighboring, paired hv and non-hvNLRs in Col-0, *cAT1G63350*, and *cAT5G38340*, and both broadly follow our observed differences between hv and non-hvNLRs (Fig. EV4). Therefore, we conclude that genomic features that distinguish hvNLR and non-hvNLRs can persist in close proximity, and likely are not driven by broader genome states but may instead be related to function and evolutionary speed.

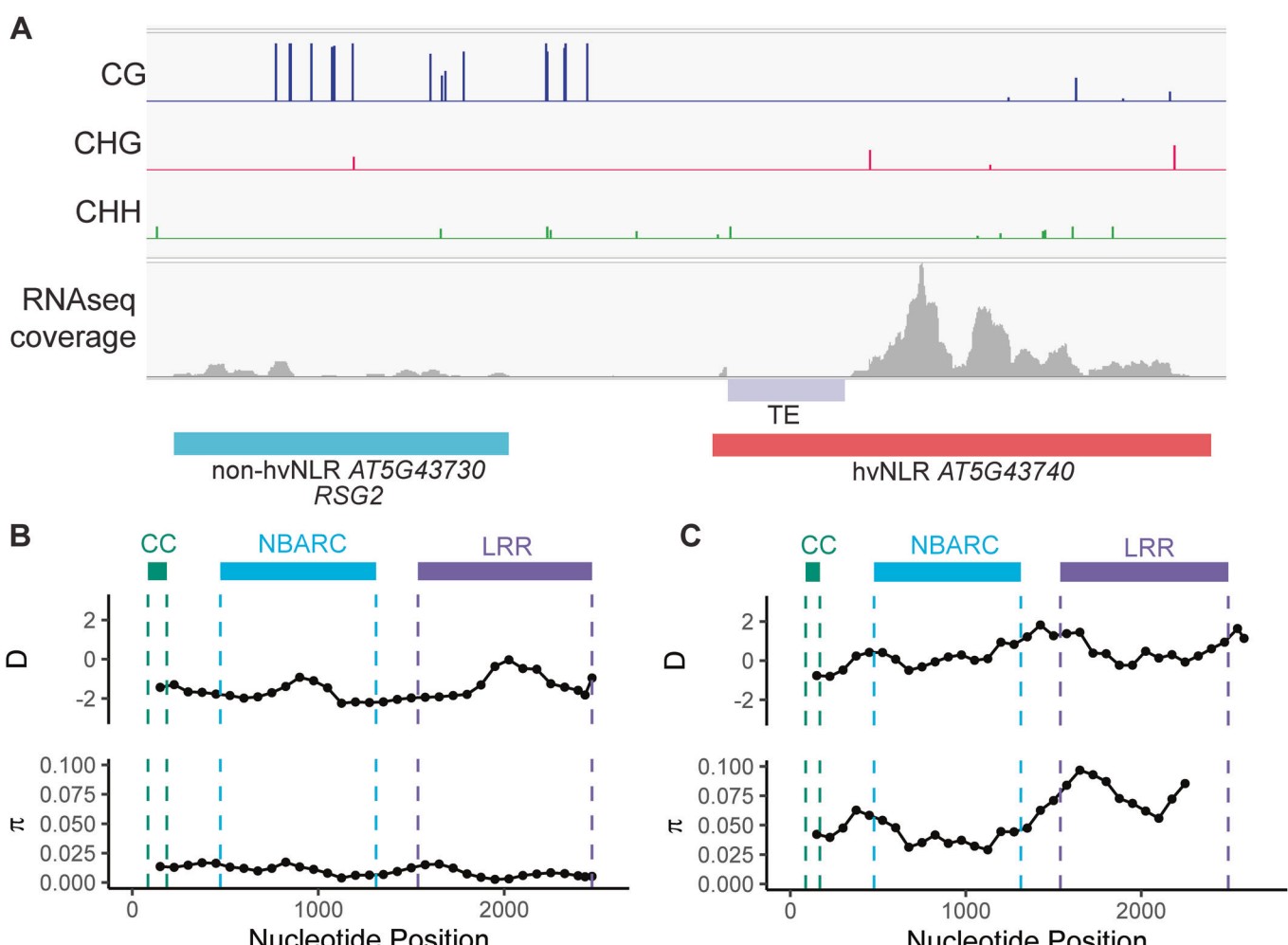

**Figure 7. An example of neighboring NLRs retaining distinct genomic and epigenomic features.**

(A) Methylation, RNAseq coverage, and TE proximity of neighboring non-hvNLR *RSG2* and hvNLR *AT5G43740*. Methylation and RNAseq coverage shown for rosette leaf tissue. (B, C) Tajima's D and nucleotide diversity across the coding sequence of *RSG2* and *AT5G43740*. Statistics were calculated on 300 bp windows with a step size of 75 bp and plotted at the nucleotide midpoint. Source data are available online for this figure.

# Discussion

The high allelic diversity of NLRs has long been appreciated, though the mechanisms that generate and maintain this diversity have remained difficult to disentangle. Taking advantage of Shannon entropy and available long-read sequencing datasets, we can delineate rapidly and slowly diversifying NLRs and begin to investigate these mechanisms through gene set comparison. Our results show that rapidly evolving NLRs have distinct genomic features from their conserved paralogs and the rest of the genome. Specifically, we found that hvNLRs are more expressed, less methylated, and closer to TEs than non-hvNLRs. Interestingly, hvNLRs are enriched across the genome in highly expressed genes, hypomethylated genes, and genes closest to TEs, while non-hvNLRs are uniformly dispersed among other genes. Differences in expression and methylation are observed across all tested shoot tissues and several important developmental and reproductive tissues, including embryo, flower, and shoot apical meristem.

Since we observed distinct genomic features between hv and non-hvNLRs, we investigated the possibility of increased mutation rate in hvNLRs through examination of nucleotide diversity and mutation probability. Synonymous substitutions are under reduced selection compared to nonsynonymous substitutions because they do not alter the amino acid sequence but are not invisible to selection due to codon bias, GC-biased gene conversion, and RNA folding stability (Martincorena et al, 2012; James et al, 2017; Wei, 2020). $\pi_S$ is therefore an imperfect measurement of mutation rate, but an elevated mutation rate of hvNLRs could result in increased $\pi_S$ and $\pi_N$ relative to non-hvNLRs, but not influence the $\pi_N/\pi_S$ ratio, as we report here (Bromham et al, 2013). We also find that hvNLRs are maintained in chromatin states associated with a higher mutation probability per base pair relative to non-hvNLRs, leading to the hypothesis that locally high mutation rate at hvNLRs contributes to the observed amino acid diversity. However, high depth quantification of de novo mutations at NLRs before selection is required to evaluate this hypothesis.

The distinct genomic features between the two NLR groups may point to mechanisms of increased mutation rate. Transcription is a source of genomic instability through the exposure of vulnerable single-stranded DNA, which is countered by targeting DNA repair machinery to actively transcribed genes through the stalling RNA polymerase or histone marks associated with actively transcribed genes (Oztas et al, 2018; Preprint: Quiroz et al, 2022). If the high transcription of hvNLRs is not accompanied by targeted DNA repair, this would result in an increased probability of mutation (Staunton et al, 2023). Methylated cytosines increase the likelihood of mutation by increasing the frequency of spontaneous deamination of cytosines (Xia et al, 2012; Weng et al, 2019; Monroe et al, 2022a). However, in *Arabidopsis*, gene body CG methylation is found preferentially in the exons of conserved, constitutively transcribed housekeeping genes, and gene body CG methylation is associated with lower polymorphism than unmethylated genes across accessions (Gaut et al, 2011; He et al, 2022; Kenchanmane Raju et al, 2023). The CG gene body methylation of non-hvNLRs may therefore be related to their low diversity through some unknown mechanism. TEs generate large effect mutations (Quadrana et al, 2019) and alter the methylation and expression landscape of surrounding genes. hvNLRs are closer to TEs and more likely to have them within their genic sequence than non-

hvNLRs, and this likely contributes to hvNLR diversification. Since highly variable-associated genomic features persist in germline tissues, any associated mutational likelihood is evolutionarily relevant.

Once generated, nucleotide diversity can be actively maintained by diversifying or balancing selection, or passively accumulate in the absence of selection. We do not observe any difference in diversifying selection between hv and non-hvNLRs using the $\pi_N/\pi_S$ metric, but hvNLRs have a significantly higher proportion of codons under pervasive and episodic diversifying selection. While hvNLRs have higher Tajima's D values than the genome average and non-hvNLRs, they are not present in the tails of the genome-wide distribution. The 5th and 95th percentiles of the empirical distribution are conservative cutoffs, and it is possible for a locus under weak selection to not be in the tail of the empirical distribution. Therefore, balancing selection may play a role in promoting hvNLR diversity but cannot be distinguished from evolution under relaxed selection using this criterion. non-hvNLRs, however, have a strong signature of purifying selection, which helps to explain their low amino acid diversity relative to hvNLRs.

Given the heterogeneous mutation rate across the *Arabidopsis* genome, it is tempting to speculate that the distinctive genomic features we observed in hvNLRs may be related to their allelic diversity. Alternatively, there might be a selection of specific features on non-hvNLRs to enhance DNA repair and inhibit other diversity-generation activities facilitating their maintenance. Our findings serve as a starting point for the investigation of the mechanisms that promote diversity generation in a subset of the plant immune receptors.

# Methods

**Reagents and tools table**

| Reagent/ resource | Reference or source | Identifier or catalog number |
|---|---|---|
| **Software** | | |
| entropy v1.3.1 | https://strimmerlab.github.io/software/entropy/ Hausser and Strimmer, 2009 | |
| Trim Galore! v0.6.6 | https://github.com/FelixKrueger/TrimGalore Babraham Bioinformatics | |
| Bismark v0.23.0 | https://github.com/FelixKrueger/Bismark Krueger and Andrews, 2011 | |
| STAR v2.7.10a | https://github.com/alexdobin/STAR Dobin et al, 2013 | |
| HTSeq v2.0.2 | https://github.com/htseq/htseq Putri et al, 2022 | |
| Polyester v1.2.0 | https://github.com/alyssafrazee/polyester Frazee et al, 2015 | |
| singscore v1.22.0 | https://bioconductor.org/packages/release/bioc/html/singscore.html Foroutan et al, 2018 | |

| Reagent/ resource | Reference or source | Identifier or catalog number |
|---|---|---|
| PAL2NAL v14 | https://www.bork.embl.de/pal2nal/ Suyama et al, 2006 | |
| EggLib v3.1.0 | https://www.egglib.org/ Siol et al, 2022 | |
| LRRpredictor v1 | https://github.com/eliza-m/ LRRpredictor_v1 Martin et al, 2020 | |
| FEL | https://hyphy.org/ Kosakovsky Pond and Frost, 2005 | |
| MEME | https://hyphy.org/ Murrell et al, 2012 | |
| vcftools v0.1.17 | https://vcftools.github.io/index.html Danecek et al, 2011 | |

## Shannon entropy

Entropy was calculated as described previously (Prigozhin and Krasileva, 2021) using the R package entropy v1.3.1 (Hausser and Strimmer, 2009), except entropy was calculated per Col-0 sequence as opposed to across the clade alignment. hvNLR and non-hvNLRs are defined as described previously (Prigozhin and Krasileva, 2021).

## DNA methylation analysis

To examine the methylation and expression of NLRs in rosette leaf tissue, we used available matched bisulfite and RNA sequencing from split Col-0 leaves (Data ref: Williams et al, 2022a, 2022b). Reads were trimmed using Trim Galore! v0.6.6 with a Phred score cutoff of 20 and Illumina adapter sequences, with a maximum trimming error rate 0.1 (Babraham Bioinformatics). Using Bismark v0.23.0, reads were mapped to the Araport11 genome, PCR duplicates were removed, and percent methylation at each cytosine was determined using the methylation extraction function (Krueger and Andrews, 2011). Cytosines with at least five reads were used for analysis, and the symmetrical cytosines within CG base pairs were averaged (Williams et al, 2022a, 2022b). The percent methylation of each CG site was averaged across each NLR gene, and across four biological replicates weighted by number of cytosines with sufficient coverage. Five hvNLR genes did not have sufficient coverage at any cytosines and were excluded from rosette leaf analysis (AT1G58807, AT1G58848, AT1G59124, AT1G59218, and AT4G26090). Single sample methylation enrichment of hvNLRs and non-hvNLRs was performed using custom permutation tests weighted for similar CG sites per gene. RPP4 and RPP7 were excluded from enrichment analysis due to their multi-context methylation.

## Gene expression analysis

RNAseq reads from four matched leaf samples (explained above) were mapped to the Araport11 genome using STAR v2.7.10a (Dobin et al, 2013) and were counted using HTSeq-count v2.0.2

(Putri et al, 2022). Counts were converted to transcripts per million (TPM) and averaged across four biological replicates, then $log_2(TPM + 1)$ transformed for visualization. NLRs are repetitive and often similar, making them difficult to sequence with short reads. To determine if any NLRs were unmappable, RNAseq reads were simulated using Polyester v1.2.0 (Frazee et al, 2015). Four NLRs were determined to be unmappable due to zero assigned read counts and were excluded from rosette tissue expression analysis (AT1G58807, AT1G58848, AT1G59124, and AT1G59218). Single sample gene set enrichment of hvNLRs and non-hvNLRs was performed on each replicate using singscore (Foroutan et al, 2018).

## Multi-tissue analysis

To examine methylation across tissues, we used available bisulfite sequencing of cauline leaf, embryo, flower bud, and distinct rosette leaf samples and processed as described above (Data ref: Williams et al, 2022a, 2022b). Two non-hvNLRs (AT4G26090 and AT5G48770) and three hvNLRs (AT1G58807, AT1G59124, and AT1G59218) did not have sufficient coverage at any cytosines and were excluded from the analysis. To examine NLR expression across tissues, we downloaded per-gene count data for 52 tissues of Arabidopsis Col-0 (Data ref: Mergner et al, 2020a, 2020b). No NLRs were determined to be unmappable with the described RNA sequencing conditions. We converted per-gene counts to TPM and $log_2 (TPM + 1)$ transformed for visualization.

## TE analysis

We determined distance to transposable elements based on the TE annotation file TAIR10_Transposable_Elements.txt and gene annotation file TAIR10_GFF3_genes.gff available from arabidopsis.org.

## Subgroup comparison

Clustered NLRs are defined as NLRs within 50 kb of an adjacent NLR (Lee and Chae, 2020). Since we were focused on physical proximity, we did not include exclusively sequence similarity-based clusters. The phylogenetic tree of all NLRs in Col-0 was generated as described previously (Prigozhin and Krasileva, 2021) with feature annotations using iTOL. Paired hvNLR and non-hvNLR neighbors were identified by 1) clusters with two NLRs, one hv and one non-hv 2) NLRs were either directly next to each other or within 2 kb of each other.

## Population genetics analysis

Protein alignments for each NLR clade were generated as described previously (Prigozhin and Krasileva, 2021) and converted to codon alignments using PAL2NAL v14 (Suyama et al, 2006). The population genetics statistics of NLRs were calculated using EggLib v3.1.0 (Siol et al, 2022). Domain-specific statistics were calculated on subsets of codon alignments using majority vote across annotations. NBARC, TIR, and CC annotations were collected from previous work (Van de Weyer et al, 2019), and LRR annotations were determined using LRRpredictor (Martin et al, 2020). Sliding window analysis was performed using 300 base pair windows with a 75 base pair step. Sites under pervasive diversifying

selection were identified using FEL (Kosakovsky Pond and Frost, 2005), and sites under episodic diversifying selection were identified using MEME (Murrell et al, 2012) using the internal branches of the phylogeny (Pond et al, 2006; Avanzato et al, 2019). Empirical distributions of population genetics statistics of coding sequences were calculated from the all sites 1001 Genomes VCF subset to the accessions used to generate the NLRome long-read dataset using vcftools v0.1.17 (Danecek et al, 2011; Alonso-Blanco et al, 2016; Van de Weyer et al, 2019).

## Data availability

No primary datasets have been generated and deposited. The intermediate datasets and computer code produced in this study are available in the following databases: Data processing pipelines and figure generation code: GitHub (https://github.com/chandlersutherland/nlr_features); Intermediate datasets: Zenodo Public Repository (https://doi.org/10.5281/zenodo.10530531) and the accompanying source data files.

## Peer review information

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

## Acknowledgements

The authors are grateful to the Krasileva Lab for the critical reading of the manuscript, and Ben Williams for his advice and comments. The summary figure was created with BioRender.com. This research used the Savio computational cluster resource provided by the Berkeley Research Computing program at the University of California, Berkeley (supported by the UC Berkeley Chancellor, Vice Chancellor for Research, and Chief Information Officer). Chandler A Sutherland has been supported by the Grace Kase-Tsujimoto Graduate Fellowship. Ksenia V Krasileva is funded by NIH Director's Award (1DP2AT011967-01), Gordon and Betty Moore Inventor Fellowship (grant number: 8802), and the Innovative Genomics Institute.

## Author contributions

**Chandler A Sutherland**: Conceptualization; Data curation; Formal analysis; Validation; Visualization; Methodology; Writing—original draft. **Daniil M Prigozhin**: Conceptualization; Data curation; Formal analysis; Supervision; Investigation; Methodology; Writing—review and editing. **J Grey Monroe**: Data curation; Formal analysis; Writing—review and editing. **Ksenia V Krasileva**: Conceptualization; Resources; Supervision; Project administration; Writing—review and editing.

## Disclosure and competing interests statement

The authors declare no competing interests.

# Expanded View Figures

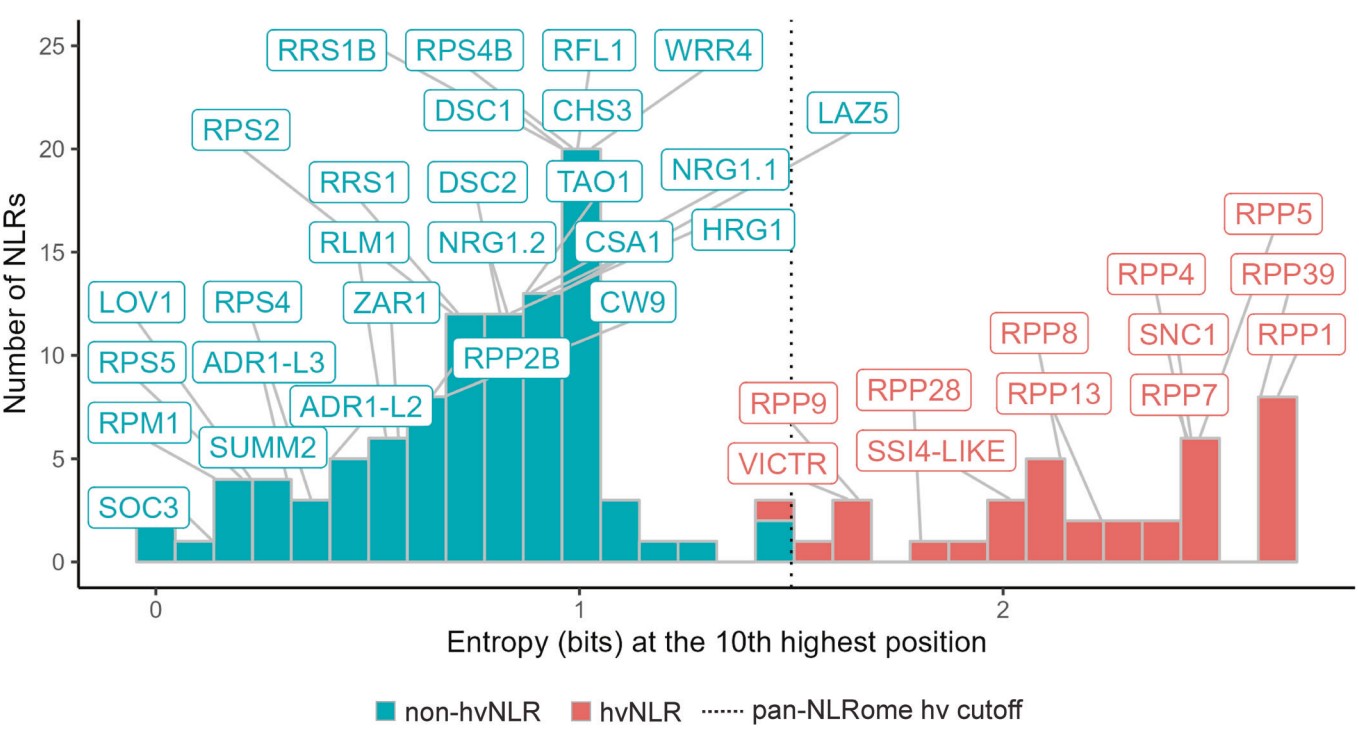

**Figure EV1.** *Arabidopsis* **Col-0 Shannon entropy at the tenth highest amino acid position.**

Distribution of NLR Shannon entropy per sequence at the tenth highest amino acid position as shown by a histogram with 30 bins. Named NLRs with previous functional characterization are labeled on the graph. The designation of hvNLR is entropy of >1.5 bits at the tenth highest position across the *Arabidopsis* NLRome, as shown by the dashed line. Source data are available online for this figure.

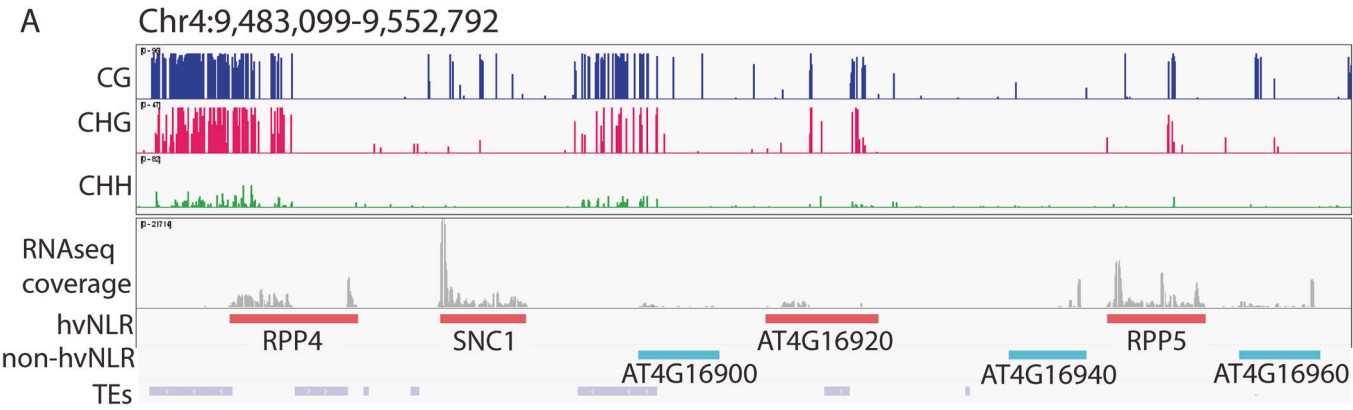

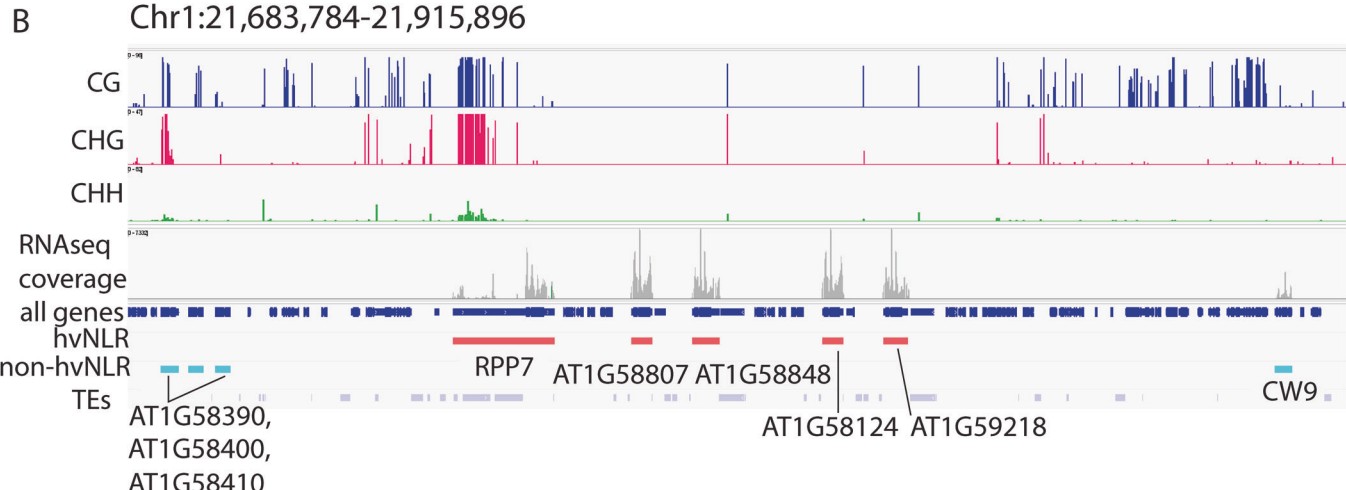

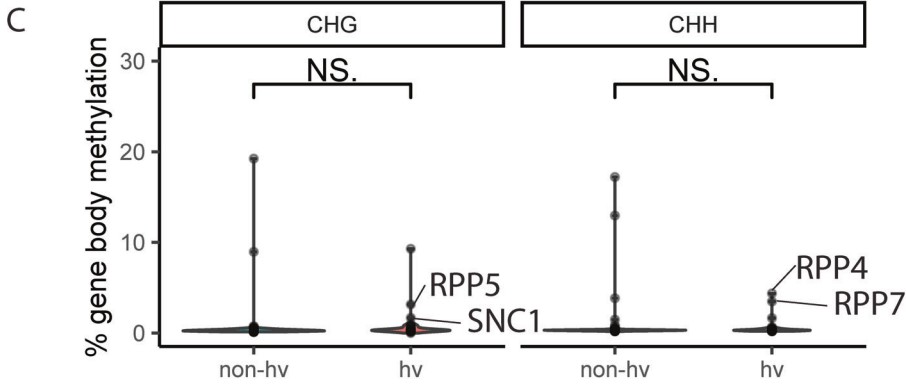

**Figure EV2. Multi-context methylation of two NLR clusters likely due to recent TE insertion.**

(A, B) Integrative Genomics Viewer screenshot of methylation, RNAseq coverage, and TE proximity of the *RPP4* and *RPP7* clusters in rosette leaf tissue. (C) % CHG and CHH gene body methylation of non-hv and hvNLRs in rosette leaf tissue. Named *RPP4* and *RPP7* cluster members are labeled. Data Information: (C) for both comparisons, $n = 97$ non-hvNLRs and $n = 35$ hvNLRs, with $n$ referring to the number of genes tested. Significance shown is the result of unpaired Wilcoxon rank-sum tests with Benjamini–Hochberg correction for multiple testing. "n.s." indicates a *P* value > 0.05. Source data are available online for this figure.

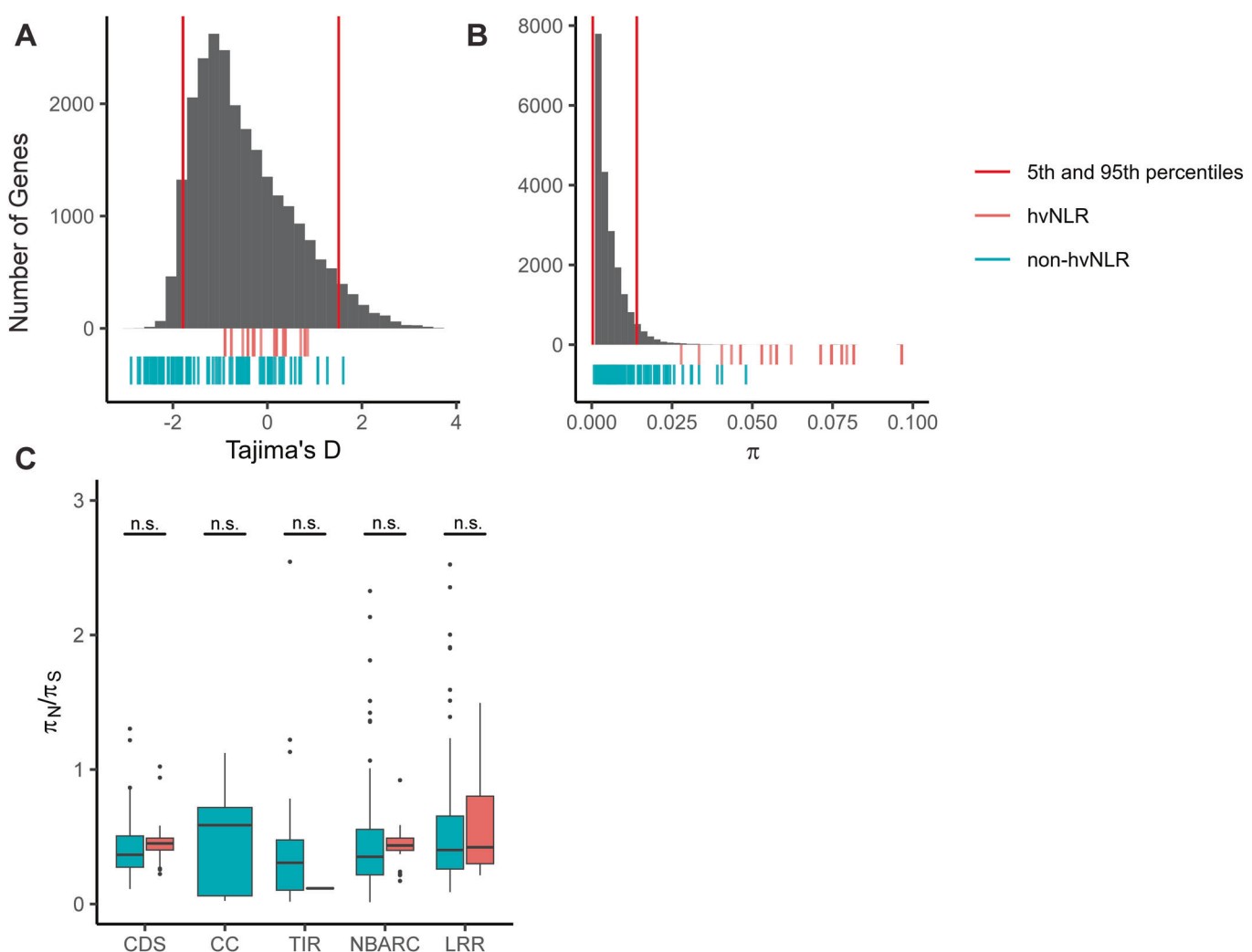

**Figure EV3. Comparison of NLR population genetics statistics to the empirical distribution; per NLR domain $\pi_N/\pi_S$.**

(A, B) Empirical distribution of Tajima's D and $\pi$ calculated on coding sequences of *Arabidopsis* shown as a histogram with 50 bins. The position of hv and non-hvNLRs shown via rug plot beneath the histogram, as well as the 5th and 95th percentiles of the distribution. (C) $\pi_N/\pi_S$ calculated per domain. Data Information: (C) horizontal black lines denote median values within each box; boxes range from the 25th to 75th percentile of each group's distribution of values; whiskers extend no further than 1.5× the interquantile range of the hinge. Data beyond the end of the whiskers are outlying points and are plotted individually. Significance shown is the result of an unpaired Wilcoxon rank-sum test with Benjamini–Hochberg correction for multiple testing. "n.s." indicates a P value > 0.05. Source data are available online for this figure.

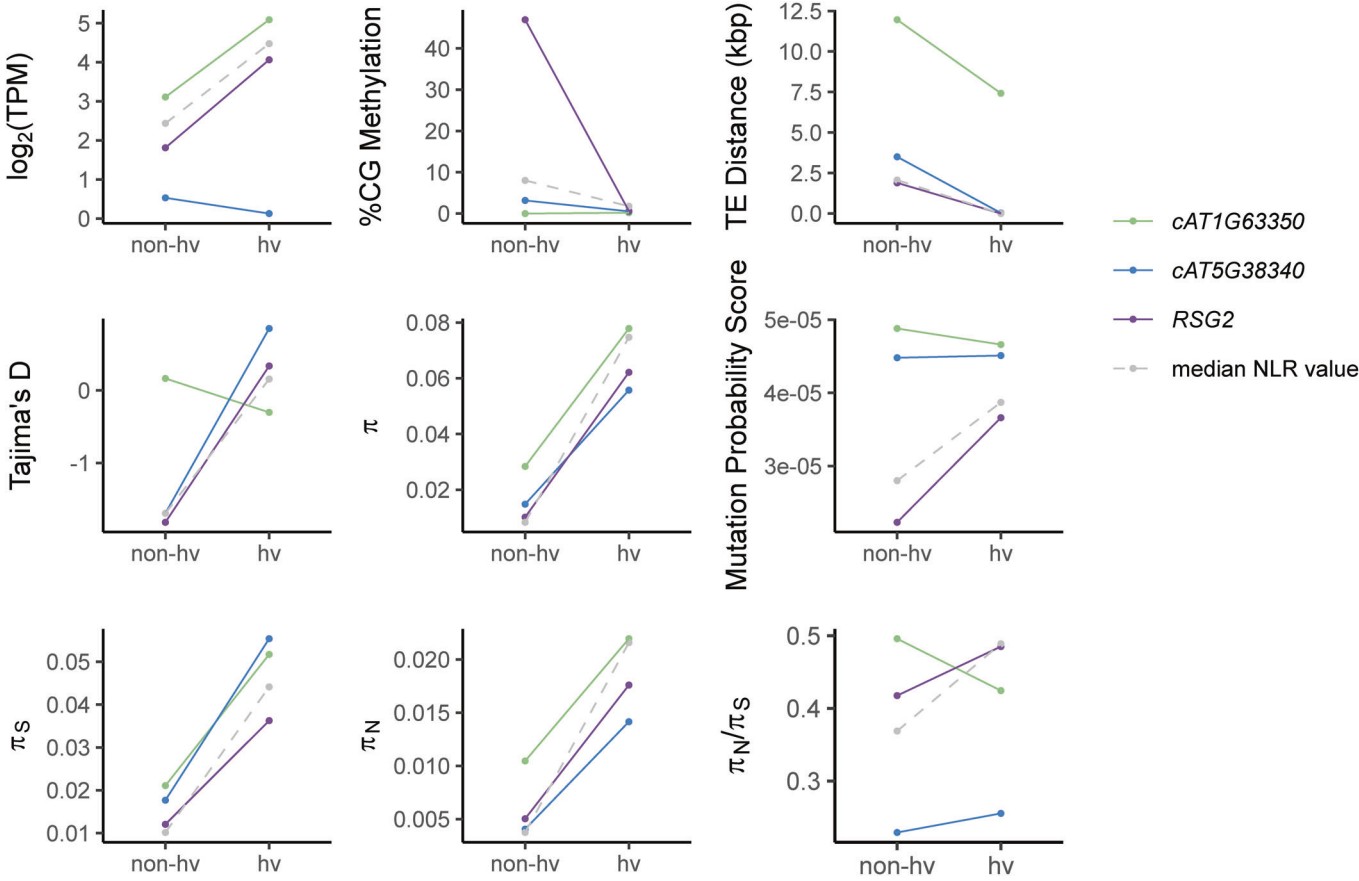

**Figure EV4. Intra-cluster comparison of neighboring hv and non-hvNLRs.**

Described genomic features and population genetics statistics are shown for hv and non-hvNLR pairs in three clusters: *RSG2, cAT1G63350,* and *cAT5G38340*. The median of all hv and non-hvNLR values is shown by a grey dotted line. Expression (log$_2$ (transcripts per million (TPM))) and % CG methylation shown is from rosette leaf tissue. Source data are available online for this figure.

