## [Peer Review File · EMBO Reports]

High allelic diversity in Arabidopsis NLRs is associated with distinct genomic features

Chandler Sutherland, Daniil Prigozhin, J.Grey Monroe, and Ksenia Krasileva

Corresponding author(s): Ksenia Krasileva (kseniak@berkeley.edu)

Review Timeline:

Submission Date:	23rd Oct 23
Editorial Decision:	16th Nov 23
Revision Received:	19th Jan 24
Editorial Decision:	16th Feb 24
Revision Received:	7th Mar 24
Accepted:	8th Mar 24

Editor: Achim Breiling

Transaction Report:

Dear Prof. Krasileva,

Thank you for the submission of your research manuscript to EMBO reports. I have now received the reports from the three referees that were asked to evaluate your study, which can be found at the end of this email.

As you will see, the referees think that the findings are of interest. However, they have several comments, concerns, and suggestions, indicating that a major revision of the manuscript is necessary to allow publication of the study in EMBO reports. As the reports are below, and all the referee concerns need to be addressed, I will not detail them here.

Given the constructive referee comments, I would like to invite you to revise your manuscript with the understanding that all referee concerns must be addressed in the revised manuscript or in a detailed point-by-point response. Acceptance of your manuscript will depend on a positive outcome of a second round of review. It is EMBO reports policy to allow a single round of revision only and acceptance of the manuscript will therefore depend on the completeness of your responses included in the next, final version of the manuscript.

- 1) a .docx formatted version of the final manuscript text (including legends for main figures, EV figures and tables), but without the figures included. Figure legends should be compiled at the end of the manuscript text.
- 2) individual production quality figure files as .eps, .tif, .jpg (one file per figure), of main figures (up to 8) and EV figures. Please upload these as separate, individual files upon re-submission.

For more details, please refer to our guide to authors:
<http://www.embopress.org/page/journal/14693178/authorguide#manuscriptpreparation>

Please consult our guide for figure preparation:
http://wol-prod-cdn.literatumonline.com/pb-assets/embo-site/EMBOPress_Figure_Guidelines_061115-1561436025777.pdf

See also the guidelines for figure legend preparation:
<https://www.embopress.org/page/journal/14693178/authorguide#figureformat>

- 3) a .docx formatted letter INCLUDING the reviewers' reports and your detailed point-by-point responses to their comments. As part of the EMBO Press transparent editorial process, the point-by-point response is part of the Review Process File (RPF), which will be published alongside your paper.
- 4) a complete author checklist, which you can download from our author guidelines (<https://www.embopress.org/page/journal/14693178/authorguide>). Please insert page numbers in the checklist to indicate where the requested information can be found in the manuscript. The completed author checklist will also be part of the RPF.

Please also follow our guidelines for the use of living organisms, and the respective reporting guidelines:
<http://www.embopress.org/page/journal/14693178/authorguide#livingorganisms>

- 5) that primary datasets produced in this study (e.g. RNA-seq, ChIP-seq, structural and array data) are deposited in an

appropriate public database. If no primary datasets have been deposited, please also state this in a dedicated section (e.g. 'No primary datasets have been generated and deposited'), see below.

The accession numbers and database should be listed in a formal "Data Availability" section (placed after Materials & Methods) that follows the model below. This is now mandatory (like the COI statement). Please note that the Data Availability Section is restricted to new primary data that are part of this study. This section is mandatory. As indicated above, if no primary datasets have been deposited, please state this in this section

Data availability

8) Regarding data quantification and statistics, please make sure that the number "n" for how many independent experiments were performed, their nature (biological versus technical replicates), the bars and error bars (e.g. SEM, SD) and the test used to calculate p-values is indicated in the respective figure legends (also for potential EV figures and all those in the final Appendix). Please also check that all the p-values are explained in the legend, and that these fit to those shown in the figure. Please provide statistical testing where applicable. Please avoid the phrase 'independent experiment', but clearly state if these were biological or technical replicates. Please also indicate (e.g. with n.s.) if testing was performed, but the differences are not significant. In case n=2, please show the data as separate datapoints without error bars and statistics. See also: <http://www.embopress.org/page/journal/14693178/authorguide#statisticalanalysis>

9) Please also note our reference format:

10) We updated our journal's competing interests policy in January 2022 and request authors to consider both actual and perceived competing interests. Please review the policy <https://www.embopress.org/competing-interests> and update your competing interests if necessary. Please name this section 'Disclosure and Competing Interests Statement' and put it after the Acknowledgements section.

11) We now use CRediT to specify the contributions of each author in the journal submission system. CRediT replaces the author contribution section. Please use the free text box to provide more detailed descriptions and do not provide your final manuscript text file with an author contributions section. See also our guide to authors:

<https://www.embopress.org/page/journal/14693178/authorguide#authorshippinguidelines>

12) We would encourage you to use 'Structured Methods', our new Materials and Methods format. According to this format, the Materials and Methods section should include a Reagents and Tools Table (listing key reagents, experimental models, software and relevant equipment and including their sources and relevant identifiers) followed by a Methods and Protocols section in which we encourage the authors to describe their methods using a step-by-step protocol format with bullet points, to facilitate the adoption of the methodologies across labs. More information on how to adhere to this format as well as downloadable

templates (.doc or .xls) for the Reagents and Tools Table can be found in our author guidelines (section 'Structured Methods'):

Please order the manuscript sections like this, using these names:

Title page - Abstract - Keywords - Introduction - Results - Discussion - Materials and Methods - Data availability section - Acknowledgements - Disclosure and Competing Interests Statement - References - Figure legends - Expanded View Figure legends

I look forward to seeing a revised version of your manuscript when it is ready. Please let me know if you have questions or comments regarding the revision.

Yours sincerely,

Referee #1:

This paper reports on a statistics-driven study of existing genome, transcriptome and methylome data on *Arabidopsis thaliana* NLR immune receptor genes providing support for defined differences in certain genome-associated features between highly variable NLRs (hvNLRs) and their low variability paralogs (non-hvNLRs). Specifically the authors are providing data supporting that hvNLRs are in chromatin states associated with higher mutation rates and show higher expression levels, less gene body methylation, and closer association with transposable elements (TEs). Their results further support diversifying selection acting at hvNLR loci, while purifying selection maintains conservation of non-hvNLRs. Overall I find this paper a valuable contribution for scientists interested in NLR gene evolution. However, the scope and impact of this study is a bit limited, as only correlations are demonstrated and no causal relationships are proven. As stated by the authors at the end of the discussion section, their findings serve as a "starting point for the investigation of the mechanisms that promote the generation of diversity among hvNLRs".

I have the following major points:

- Please rephrase the statement made in the results section about the data shown in fig 2A. "...hvNLRs are expressed significantly higher than non hv NLRs". It seems to me that there is at least one non-hvNLR that is expressed higher than any of the hvNLRs. I assume that the authors are referring to the distribution rather than making a general comment about all members of each group.

Besides this, given the high abundance of transcriptomics data that are available for *A. thaliana*, the conclusion that hvNLRs tend to be expressed at higher levels could have been further supported by using additional data sets, perhaps including additional tissue types. Would higher mutation rates associated with high levels of expression affect the germ line if only observed in rosette leaf tissue? The same applies to the possible effects of cytosine-methylation associated effects.

- If I don't understand this wrong, in Figure 3A observations are reported for "clustered NLRs" in comparison to "all NLRs". Please define what is meant by "clustered".

Based on this analysis the following conclusion was drawn: "the highly variable status of NLRs is not dependent on cluster membership". I think this statement would be better supported if "clustered NLRs" were compared to "non-clustered NLRs" instead of "all NLRs". A large number of NLRs are clustered. If the vast majority of "all NLRs" are "clustered", then their conclusion may be wrong. Please state how many of "all NLRs" considered are "clustered".

- As this manuscript is submitted to a journal with a wide-audience, it may have been good to explain a little bit more about some of the statistics used here. E.g. what is "Tajima's D"?

Referee #2:

In this work, Sutherland and colleagues explore various associations between highly variable NLRs (hvNLRs) and their expression, methylation, and physical proximity to TEs in the *Arabidopsis thaliana* Col-0 accession. While many of the general findings align with prior expectations, the study does highlight some novel specific correlations.

The authors emphasize that this investigation serves as a foundational step towards understanding the genomic mechanisms

fostering hvNLR diversification, which I agree with. Their assessment of the correlations is meticulous, and they've made substantial efforts to minimize potential confounding factors. The figures provided in the study are both informative and well-illustrated.

However, questions remain regarding the applicability of these findings to other *Arabidopsis thaliana* ecotypes or related species. It should be underscored that this research primarily focuses on the Col-0 accession. Despite this specificity, the title, especially its emphasis on "Intraspecies allelic diversity," gives the impression that NLRs from multiple accessions were analyzed, which can be misleading.

General comments:

-Consider incorporating line numbers for ease of reference.

Introduction

-First paragraph lacks appropriate references

- "After binding of a pathogen target to the LRR domain" confusing as not always the case/unknown

- "Plant NLRs are differentiated into three anciently diverged classes based on their N-terminal domains" This is based on phylogeny, the N-term follows this phylogeny

- "NLRs are organized into clusters more often than other genes, which can asymmetrically drive NLR expansion and diversification through unequal crossing over and gene conversion (Michelmore and Meyers, 1998; Lee and Chae, 2020)" tandem duplications could be mentioned

- "The NLR gene family includes the most polymorphic loci and contains the highest frequency of major effect mutations in the *Arabidopsis* genome (Gan et al., 2011)." Mentioned in Clark et al., 2007 doi: 10.1126/science.1138632.

Results:

-In the section preceding Figure 2, where "Dangerous mix genes" are discussed, consider citing Bomblies (2007) and Chae (2014).

-NBARC should be spelled out as nucleotide binding adaptor shared by APAF-1, certain R gene products and CED4 in the first instance, instead of as nucleotide-binding domain. This is to indicate that it refers to the whole NBARC and to distinguish it from the nucleotide-binding domain (NBD), which is one of the three sub-domains that makes up the NBARC (NBD, HD1 and WHD/ARC1 and ARC2 depending on the nomenclature).

-When discussing the Col-0 RPP7 TE insertion in the paragraph associated with Figure 2B, reference Tsuchiya & Elgem (2013) for comprehensive context.

-Supp fig. 1: What is the dotted line? Why is there a HV and a non-HV in the same bin? Is this panNLRome based, what is difference to Fig.1?

- "dangerous mix genes" needs to be described first

- "When we ranked all protein coding *Arabidopsis* genes based on their expression level, we observed that hvNLRs are enriched in the most expressed genes in each leaf sample" This is shown in the lower panel?

-Supp Fig.2: "which we rarely observed in other NLRs" can this be quantified

-Fig. 2C: not bold in text

-Fig. 3: do HV more frequently cluster with HV, or how are the clusters composed?. For Figure 6, it might be beneficial to include additional examples of both hvNLRs and non-hvNLRs to provide a more comprehensive overview.

Methods

-Htseq counts reference is missing/wrong

-Provide github for used packages e.g. ComplexUpset

-Foroutan et al., 2018 is mentioned in main text, could better be mentioned in the methods part

Referee #3:

In their study, Sutherland and colleagues use publicly available and paired RNA-seq and epigenomic data to show that particular plant immunity-related genes (NLRs), which are known to have high intra-specific diversity, are associated with certain genomic and epigenomic features. Throughout their study, the authors compare pre-defined sets of highly variable (hv) and less variable (non-hv) NLRs in the model plant *A. thaliana* with respect to various genomic and epigenomic features.

Among other things, the authors show that hv-NLRs are on average closer to transposable elements, are more highly expressed, and have lower gene body methylation. Previous studies have shown that plant immunity-related genes are enriched for TEs; what is new in this study is that the authors can show that this effect is driven by hvNLRs in particular. Using population genetics approaches, the authors show that non-hv-NLRs are under purifying selection whereas hv-NLRs seem to underlie higher mutation rates and/or less frequent repair.

All in all, I found the study to be very well designed and the results to be presented in a very clear and concise way. The results should provide an interesting starting point for future research into the evolution and functional diversification of plant immunity-related genes.

I have only one slightly larger criticism: in the last part of their manuscript, the authors make the point that the overall configuration of the local genomic region that contains the NLR is not the decisive factor. To do so, they employ a comparison between two adjacent NLR genes, one a hv-NLR, the other a non-hv-NLR. Compelling as this is, it is based on one single locus. I would recommend that the authors search for more of these hv/non-hv neighbor pairs to strengthen this very interesting point.

Minor comments:

- Figure 2 and related results: the sample size between the two groups (hv and non-hv) is drastically different. To make a fair statistical comparison, one should randomly and repeatedly subsample the non-hv population.
- Figure 3: as far as I can tell, the color code for hv and non-hvNLRs is not provided.

Referee #1:

This paper reports on a statistics-driven study of existing genome, transcriptome and methylome data on *Arabidopsis thaliana* NLR immune receptor genes providing support for defined differences in certain genome-associated features between highly variable NLRs (hvNLRs) and their low variability paralogs (non-hvNLRs). Specifically the authors are providing data supporting that hvNLRs are in chromatin states associated with higher mutation rates and show higher expression levels, less gene body methylation, and closer association with transposable elements (TEs). Their results further support diversifying selection acting at hvNLR loci, while purifying selection maintains conservation of non-hvNLRs. Overall, I find this paper a valuable contribution for scientists interested in NLR gene evolution. However, the scope and impact of this study is a bit limited, as only correlations are demonstrated, and no causal relationships are proven. As stated by the authors at the end of the discussion section, their findings serve as a "starting point for the investigation of the mechanisms that promote the generation of diversity among hvNLRs".

I have the following major points:

- Please rephrase the statement made in the results section about the data shown in fig 2A. "...hvNLRs are expressed significantly higher than non hv NLRs". It seems to me that there is at least one non-hvNLR that is expressed higher than any of the hvNLRs. I assume that the authors are referring to the distribution rather than making a general comment about all members of each group.

> We thank the reviewer for this point and have clarified our wording. We are referring to the hvNLRs as a set compared to non hvNLRs throughout the paper. We now more explicitly report the result of the unpaired Wilcoxon rank-sum test, which is testing for significant differences in distributions, using the following language: "We found that the distribution of hvNLR expression is significantly higher than non-hvNLRs" (lines 123-124) or explicitly refer to the test as a difference in groups "In addition, the hvNLRs gene set is significantly less CG gene body methylated than non-hvNLRs" (lines 128-129) and adopt this language throughout the manuscript when describing this statistical test.

Besides this, given the high abundance of transcriptomics data that are available for *A. thaliana*, the conclusion that hvNLRs tend to be expressed at higher levels could have been further supported by using additional data sets, perhaps including additional tissue types. Would higher mutation rates associated with high levels of expression affect the germ line if only observed in rosette leaf tissue? The same applies to the possible effects of cytosine-methylation associated effects.

> We thank the reviewer for this suggestion and have repeated our comparison of hv and non-hvNLR expression in 52 tissue types and of methylation in 4 additional tissue types, now included as Figure 4. Our observed expression trends are consistent in reproductive tissues including all stages of flower development, 4 of 5 measured stages of embryo development, all silique and fruit tissues tested. The trends are different in seed tissue and root tissue. Methylation associations are consistent across all available tissues.

- If I don't understand this wrong, in Figure 3A observations are reported for "clustered NLRs" in comparison to "all NLRs". Please define what is meant by "clustered".

> We added our explicit definition of “cluster” to the main text (Lines 152-153) and methods (Line 376) instead of only citing the defining paper. We are using a previously reported gene cluster distance designation of 50kb to the nearest NLR (Lee & Chae, 2020).

Based on this analysis the following conclusion was drawn: "the highly variable status of NLRs is not dependent on cluster membership". I think this statement would be better supported if "clustered NLRs" were compared to "non-clustered NLRs" instead of "all NLRs". A large number of NLRs are clustered. If the vast majority of "all NLRs" are "clustered", then their conclusion may be wrong. Please state how many of "all NLRs" considered are "clustered".

> We thank the reviewer for their suggestion. To address the valuable point of the majority of NLRs being clustered, we now explicitly show singletons in **Fig 3A** and clarify the sample sizes of each subset. We have now added explicit statistical testing between the hvNLR subsets and between non-hvNLRs subsets and found them to be not significantly different. Therefore, we are confident that cluster status and N-terminal domain are not confounding factors in our observed feature associations.

- As this manuscript is submitted to a journal with a wide audience, it may have been good to explain a little bit more about some of the statistics used here. E.g. what is "Tajima's D"?

> We have added additional clarification of the population genetics terms and statistics used, “D is a site frequency spectrum-based statistic that tests for selection by comparing the difference between the average number of nucleotide differences and the total number of segregating sites to the neutral expectation, while π measures the degree of polymorphism within a population by the average pairwise differences per site. In comparison to the rest of the genome, these statistics can be used to test for balancing selection (Schmid *et al*, 2005)” (Lines 196-201).

Referee #2:

In this work, Sutherland and colleagues explore various associations between highly variable NLRs (hvNLRs) and their expression, methylation, and physical proximity to TEs in the *Arabidopsis thaliana* Col-0 accession. While many of the general findings align with prior expectations, the study does highlight some novel specific correlations.

The authors emphasize that this investigation serves as a foundational step towards understanding the genomic mechanisms fostering hvNLR diversification, which I agree with. Their assessment of the correlations is meticulous, and they've made substantial efforts to minimize potential confounding factors. The figures provided in the study are both informative and well-illustrated.

However, questions remain regarding the applicability of these findings to other *Arabidopsis thaliana* ecotypes or related species. It should be underscored that this research primarily focuses on the Col-0 accession. Despite this specificity, the title, especially its emphasis on "Intraspecies allelic diversity," gives the impression that NLRs from multiple accessions were analyzed, which can be misleading.

> We thank the reviewer for this comment and have considered it extensively. We use the phrase “intraspecies allelic diversity” to describe hvNLR status and our reported population genetics statistics, which are calculated across accessions. We want to emphasize our core result of the paper in the title, which is a reflection of speeds of evolution observed at the intraspecies level on the genomic features of a single accession. However, we understand that our description of the data used in Figures 2 and 3 is unclear and potentially misleading. We now introduce the use of a single accession in the results of Figure 1, stating “To examine the relationships between population level diversity and genomic features of a single accession, we plotted Shannon entropy in reference to each NLR in Col-0” (Line 112-113). We have also added the Col-0 accession name to the results section of Figure 2 and emphasize the use of a single plant: “To compare the expression and methylation status of hv and non-hvNLRs within an individual plant, we examined available paired whole genome bisulfite and RNA sequencing generated from the same Col-0 rosette leaf” (Line 120-123). In our new analysis of multiple tissue types, we continue to explicitly denote they are derived from Col-0.

> We chose to perform this analysis only in Col-0 due to the requirement of long read, *de novo* assembled genomes for analysis of NLR features. With future *Arabidopsis* sequencing projects, the feature analysis could be repeated across the species, but we are confident in these reported trends due to our additional tissue analysis. We have also observed the same trends across the pangenome of maize (work in preparation).

General comments:

-Consider incorporating line numbers for ease of reference.

>We have added line numbers and refer to them throughout this document.

Introduction

-First paragraph lacks appropriate references

>We have added references to both primary and secondary literature to add context to the importance of population-level receptor diversity in host immune system durability.

-"After binding of a pathogen target to the LRR domain" confusing as not always the case/unknown

>We thank the reviewer for pointing out our error, and have changed the language of how the LRR domain works to allow for the uncertainty of the mechanism: “a leucine-rich repeat (LRR) domain involved in direct or indirect recognition of pathogens” (Lines 46-47).

-"Plant NLRs are differentiated into three anciently diverged classes based on their N-terminal domains" This is based on phylogeny, the N-term follows this phylogeny

> We reworded this sentence to place emphasis on the phylogenetic classification and included references describing the phylogenetic analysis. “NLRs are grouped into three anciently diverged classes based on their N-terminal domains: coiled-coil (CC) NLRs (CNL), RPW8-like coiled-coil NLRs (RNL), and Toll/Interleukin-1 receptor (TIR) NLRs (TNLs) (Shao *et al*, 2016; Tamborski and Krasileva 2020).” (Lines 47-50).

-"NLRs are organized into clusters more often than other genes, which can asymmetrically drive NLR expansion and diversification through unequal crossing over and gene conversion (Michelmore and Meyers, 1998; Lee and Chae, 2020)" tandem duplications could be mentioned

> At the reviewer’s suggestion we have added mention of tandem duplication and a citation of the description of tandem duplications in the evolution of RPP5: “NLRs are in close proximity to each other in genomes and are organized into clusters more often than other genes. This proximity can asymmetrically drive NLR expansion and diversification through tandem duplication, unequal crossing over, and gene conversion (Parker *et al*, 1997; Michelmore & Meyers, 1998; Lee & Chae, 2020)” (Lines 55-59).

-"The NLR gene family includes the most polymorphic loci and contains the highest frequency of major effect mutations in the Arabidopsis genome (Gan et al., 2011)." Mentioned in Clark et al., 2007 doi: 10.1126/science.1138632.

>We thank the reviewer for this citation recommendation and have included it in the manuscript (line 63).

Results:

-In the section preceding Figure 2, where "Dangerous mix genes" are discussed, consider citing Bomblies (2007) and Chae (2014).

> We thank the reviewer for the citation suggestions and have included them in the manuscript as well as a description of dangerous mix genes: "In addition, hvNLRs include all currently known dangerous mix genes that are responsible for hybrid incompatibility across Arabidopsis accessions (Bomblies *et al*, 2007; Chae *et al*, 2014)." (Lines 115-117).

-NBARC should be spelled out as nucleotide binding adaptor shared by APAF-1, certain R gene products and CED4 in the first instance, instead of as nucleotide-binding domain. This is to indicate that it refers to the whole NBARC and to distinguish it from the nucleotide-binding domain (NBD), which is one of the three sub-domains that makes up the NBARC (NBD, HD1 and WHD/ARC1 and ARC2 depending on the nomenclature).

>We thank the reviewer for this important point, and have clarified our use of the NBARC acronym in the introduction: "NLRs have a modular domain structure, with a variable N-terminal domain involved in downstream signaling, a central nucleotide-binding domain shared by APAF-1, various other plant immune proteins, and CED4 (NBARC), and a leucine-rich repeat (LRR) domain involved in direct or indirect recognition of pathogens" (Lines 44-48).

-When discussing the Col-0 RPP7 TE insertion in the paragraph associated with Figure 2B, reference Tsuchiya & Elgem (2013) for comprehensive context.

> We thank the reviewer for this recommendation and have included it in the manuscript (Line 137).

-Supp fig. 1: What is the dotted line? Why is there a HV and a non-HV in the same bin? Is this panNLRome based, what is difference to Fig.1?

> The dotted line in Supplemental Figure 1 (Now Fig EV1) represents the definition of an hvNLR as entropy > 1.5 bits at the tenth highest amino acid position across the NLRome. This

was described in the figure legend, but we have now included this in the plot. The difference between Fig EV1 and Fig 1 is the choice of x axis. Mean per-gene Shannon entropy, as shown in Fig 1, is easier to understand than entropy at the tenth highest amino acid position, but we wanted to include both to show that the hv designation does not depend exclusively on the threshold chosen to define it. The delineation of hvNLRs is a panNLRome metric and described in Prigozhin and Krasileva 2021. Because we are focusing on Col-0, we calculated entropy per Col-0 sequence as opposed to across the alignment. That is why there is an HV in the non-HV bin, is that the allelic diversity definition is based on pan-genome, and the data shown here is in reference to gene identifiers in Col-0. We have included this information explicitly in the methods to clarify the difference in the results reported in this paper and previously (Lines 333-337).

- "dangerous mix genes" needs to be described first

>We agree, and please see our earlier response to incorporating description of dangerous mix genes.

- "When we ranked all protein coding Arabidopsis genes based on their expression level, we observed that hvNLRs are enriched in the most expressed genes in each leaf sample" This is shown in the lower panel?

>Yes, and we have updated our figure panel lettering to make this explicit.

- Supp Fig.2: "which we rarely observed in other NLRs" can this be quantified

> We thank the reviewer for this suggestion and have added the median % CHH and %CHG gene body methylation to the manuscript text (Line 155) and panel C to Fig EV 2 that shows the distribution of hv and non-hvNLR % CHH and CHG methylation.

- Fig. 2C: not bold in text

>We have fixed this.

- Fig. 3: do HV more frequently cluster with HV, or how are the clusters composed?. For Figure 6, it might be beneficial to include additional examples of both hvNLRs and non-hvNLRs to provide a more comprehensive overview.

> There are 6 clusters with mixed hv and non-hvNLR membership in Col-0, including the RPP7 and RPP4/5 clusters shown in Extended View Figure 2, and the RSG2 cluster shown in Figure 7 (formerly Fig 6). Of the 22 clustered hvNLRs, 14 are in clusters with non-hvNLRs, and 8 are in hv-exclusive clusters. For within-cluster comparison, we focus on clusters composed of one

hvNLR and one non-hvNLR directly next to (or within 2kb) of each other to allow for unambiguous comparison. There are three clusters in Col-0 which fit these criteria: the currently displayed CNL *RSG2* cluster, the CNL cluster *cAT1G63350*, and the TNL cluster *cAT5G38340*. We thank the reviewer for pointing out the need for more examples, and we have now included the feature values and population genetics statistics for all three paired clusters as EV Fig 4. While not every within-cluster comparison follows the median hv vs non-hvNLR comparison, the trends broadly hold. Accordingly, we have updated our description of the results of Figure 7 to reflect several examples, but too small of a sample size to make conclusive statements about mixed cluster features (Lines 255-268).

Methods

-Htseq counts reference is missing/wrong

-Provide github for used packages e.g. ComplexUpset

-Foroutan et al., 2018 is mentioned in main text, could better be mentioned in the methods part

> We have fixed the HTseq counts reference (Putri *et al*, 2022) and moved the singscore reference to the methods. At the suggestion of this referee and the editor, we have listed all software used in our analysis in a reagents and tools table, including the reference and github or otherwise stable source code link. The references and versions are repeated in the methods text.

Referee #3:

In their study, Sutherland and colleagues use publicly available and paired RNA-seq and epigenomic data to show that particular plant immunity-related genes (NLRs), which are known to have high intra-specific diversity, are associated with certain genomic and epigenomic features. Throughout their study, the authors compare pre-defined sets of highly variable (hv) and less variable (non-hv) NLRs in the model plant *A. thaliana* with respect to various genomic and epigenomic features.

Among other things, the authors show that hv-NLRs are on average closer to transposable elements, are more highly expressed, and have lower gene body methylation. Previous studies have shown that plant immunity-related genes are enriched for TEs; what is new in this study is that the authors can show that this effect is driven by hvNLRs in particular. Using population genetics approaches, the authors show that non-hv-NLRs are under purifying selection whereas hv-NLRs seem to underlie higher mutation rates and/or less frequent repair.

All in all, I found the study to be very well designed and the results to be presented in a very clear and concise way. The results should provide an interesting starting point for future research into the evolution and functional diversification of plant immunity-related genes.

I have only one slightly larger criticism: in the last part of their manuscript, the authors make the point that the overall configuration of the local genomic region that contains the NLR is not the decisive factor. To do so, they employ a comparison between two adjacent NLR genes, one a hv-NLR, the other a non-hv-NLR. Compelling as this is, it is based on one single locus. I would recommend that the authors search for more of these hv/non-hv neighbor pairs to strengthen this very interesting point.

> There are 6 clusters with mixed hv and non-hvNLR membership in Col-0, including the RPP7 and RPP4/5 clusters shown in Extended View Figure 2, and the RSG2 cluster shown in Figure 7 (formerly Fig 6). Of the 22 clustered hvNLRs, 14 are in clusters with non-hvNLRs, and 8 are in hv-exclusive clusters. For within-cluster comparison, we focus on clusters composed of one hvNLR and one non-hvNLR directly next to (or within 2kb) of each other to allow for unambiguous comparison. There are three clusters in Col-0 which fit these criteria: the currently displayed CNL *RSG2* cluster, the CNL cluster *cAT1G63350*, and the TNL cluster *cAT5G38340*. We thank the reviewer for pointing out the need for more examples, and we have now included the feature values and population genetics statistics for all three paired clusters as EV Fig 4. While not every within-cluster comparison follows the median hv vs non-hvNLR comparison, the trends broadly hold. Accordingly, we have updated our description of the results of Figure 7 to

reflect several examples, but too small of a sample size to make conclusive statements about mixed cluster features (Lines 255-268).

Minor comments:

- Figure 2 and related results: the sample size between the two groups (hv and non-hv) is drastically different. To make a fair statistical comparison, one should randomly and repeatedly subsample the non-hv population.

>We thank the reviewer for their concern, and we shared it in our initial experimental design. We consulted with experts from the UC Berkeley department of Statistics, and chose the unpaired Wilcoxon rank sum test (aka Mann-Whitney U test) for our hv vs non-hvNLR statistical comparisons throughout the manuscript because it is applicable to non-parametric distributions and appropriate for comparisons of different sample sizes (Mann & Whitney, 1947). We prefer to use this statistic that captures the entire distribution rather than down sample, though we appreciate and understand the concern.

- Figure 3: as far as I can tell, the color code for hv and non-hvNLRs is not provided.

>We have added a color code to Figure 3A.

I also wanted to comment on something reviewer2 mentioned: the fact that this is all about Col-0 ecotype. I went back to the manuscript after reading this comment, and it is true that there seems to be an ambiguity here. When reviewing, I was under the assumption that the authors refer to allelic diversity across the *A. thaliana* population (using e.g. the 1001 genomes resource), but always referring to Col-0 as the reference sequence. In which case it would indeed be intra-specific variability. However, when revisiting, I noticed that they did not make this clear. This is something that definitely needs to be addressed or clarified.

> We thank the reviewer for this comment and have considered it extensively. We use the phrase “intraspecies allelic diversity” to describe hvNLR status and our reported population genetics statistics, which are calculated across accessions. We want to emphasize our core result of the paper in the title, which is a reflection of speeds of evolution observed at the intraspecies level on the genomic features of a single accession. However, we understand that our description of the data used in Figures 2 and 3 is unclear and potentially misleading. We now introduce the use of a single accession in the results of Figure 1, stating “To examine the relationships between population level diversity and genomic features of a single accession, we plotted Shannon entropy in reference to each NLR in Col-0” (Line 112-113). We have also added the Col-0 accession name to the results section of Figure 2 and emphasize the use of a single plant: “To compare the expression and methylation status of hv and non-hvNLRs within an individual plant,

we examined available paired whole genome bisulfite and RNA sequencing generated from the same Col-0 rosette leaf” (Line 120-123). In our new analysis of multiple tissue types, we continue to explicitly denote they are derived from Col-0.

> We chose to perform this analysis only in Col-0 due to the requirement of long read, *de novo* assembled genomes for analysis of NLR features. With future *Arabidopsis* sequencing projects, the feature analysis could be repeated across the species, but we are confident in these reported trends due to our additional tissue analysis. We have also observed the same trends across the pangenome of maize (work in preparation).

Dear Prof. Krasileva,

Thank you for the submission of your revised manuscript to our editorial offices. I have now received the reports from the referees that I asked to re-evaluate your study, you will find below. As you will see, referees #2 and #3 now fully supports the publication of the study in EMBO reports. Referee #1 states, although almost all of his/her points were adequately addressed, that the scope and impact of this study is limited and that s/he is not convinced that the paper is suitable for a wider readership. However, considering that the other referees have not brought up such concerns, and after further editorial assessment, I decided to proceed with the manuscript.

Before formal acceptance, I have these editorial requests I ask you to address in a final revised manuscript:

- Please provide a final title with not more than 100 characters (including spaces).

- Please remove the words 'Title Page' and 'Authors' from the title page, as well as the ORCID IDs. Please link the ORCID IDs to the author profiles in our submission system (if not already done). Please find instructions on how to link the ORCID ID to the account in our manuscript tracking system in our Author guidelines:

<http://www.embopress.org/page/journal/14693178/authorguide#authorshipguidelines>

- We plan to publish your manuscript in the Report format (as also indicated by you in the submission system). For this, there is a limit of 5 main and 5 EV figures. Please combine panel or rearrange the figure in a way to have 5 final main and 5 final EV figures. Please also update any call-outs that might be affected by these changes. Please also re-label the source data accordingly. Moreover, for a Scientific Report we require that results and discussion sections are combined in a single chapter called "Results & Discussion". Please do this for your manuscript. For more details, please refer to our guide to authors:

<http://www.embopress.org/page/journal/14693178/authorguide#researcharticleguide>

- Please make sure that the number "n" for how many independent experiments were performed, their nature (biological versus technical replicates), the bars and error bars (e.g. SEM, SD) and the test used to calculate p-values is indicated in the respective figure legends (for main and EV figures) of the final revised manuscript. Please also check that all the p-values are explained in the legend, and that these fit to those shown in the figure. Please provide statistical testing where applicable. Please avoid the phrase 'independent experiment', but clearly state if these were biological or technical replicates. Please also indicate (e.g. with n.s.) if testing was performed, but the differences are not significant. In case n=2, please show the data as separate datapoints without error bars and statistics. See also:

<http://www.embopress.org/page/journal/14693178/authorguide#statisticalanalysis>

If n<5, please show single datapoints for diagrams. Moreover:

- Please indicate the statistical test used for data analysis in the legends of figures 2d-f; 5a; EV 2c.

- Please note that in figures 6a, c-d; EV 3c; there is a mismatch between the annotated p values in the figure legend and the annotated p values in the figure file that should be corrected.

- Please define the box plots in terms of minima, maxima, centre, bounds of box and whiskers, and percentile in the legend of figure EV 3c.

- Please note that information related to n is missing in the legends of figures 4a, c; 6a, c-d; EV 3c.

- Although 'n' is provided, please describe the nature of entity for 'n' in the legends of figures 2a-c; 3a; 5a.

- Please remove the reagents and tools table from the main manuscript text file. I have attached templates for that in word or excel format. Please upload the filled in table to the manuscript tracking system as 'Reagent Table' file. Please also adjust any callouts to this table. The example linked below shows how the table will display in the published article and includes examples of the type of information that should be provided for the different categories of reagents and tools. Please list your reagents/tools using the categories provided in the template and do not add additional subheadings to the table. Reagents/tools that do not fit in any of the specific categories can be listed under "Other":

https://www.embopress.org/pb%2Dassets/embo-site/msb_177951_sample_FINAL.pdf

- In the manuscript text there are these callouts for data references: Data ref: Williams et al, 2022, Data ref: Mergner et al, 2020, Data ref: Monroe et al, 2022. However, these are only listed in the reference list as journal articles. We would need an additional data references each for these (below the citation of the related paper). Data citations must be labeled with "[DATASET]" in the reference list and must provide the database name, accession number/identifiers and a resolvable link to the landing page from which the data can be accessed at the end of the reference. Further instructions are available at:

- Please make sure that all the funding information is also entered into the online submission system and that it is complete and similar to the one in the acknowledgement section of the manuscript text file. Presently, a grant (?) 'Grace Kase-Tsujimoto Graduate Fellowship' is only mentioned in the acknowledgements.

- Please provide/upload the source data for the final EV figures zipped up into one folder.

In addition, I would need from you:

Best,

Referee #1:

The manuscript has been substantially improved and almost all of my critique points were adequately addressed. However, I am still not convinced that this paper is suitable for a wide readership and will be of high impact. As I had stated in my previous review " the scope and impact of this study is a bit limited, as only correlations are demonstrated, and no causal relationships are proven." My opinion in this respect has not changed.

Referee #2:

The authors have done a great job integrating feedback from all reviewers as far as I am concerned. The manuscript was already of high quality to begin with and is now much improved. I have no further comments and look forward to seeing this published.

Referee #3:

The authors have adequately addressed the points that I had raised during the initial review; I have no further comments or suggestions.

All editorial and formatting issues were resolved by the authors.

Prof. Ksenia Krasileva
University of California, Berkeley
Plant and Microbial Biology
Berkeley, CA 94720
United States

Dear Prof. Krasileva,

I am very pleased to accept your manuscript for publication in the next available issue of EMBO reports. Thank you for your contribution to our journal.

Yours sincerely,
